# ON THE EXPECTED RUNNING TIME OF NONCONVEX OPTIMIZATION WITH EARLY STOPPING

## ABSTRACT

This work examines the convergence of stochastic gradient algorithms that use early stopping based on a validation function, wherein optimization ends when the magnitude of a validation function gradient drops below a threshold. We derive conditions that guarantee this stopping rule is well-defined and analyze the expected number of iterations and gradient evaluations needed to meet this criteria. The guarantee accounts for the distance between the training and validation sets, measured with the Wasserstein distance. We develop the approach for stochastic gradient descent (SGD), allowing for biased update directions subject to a Lyapunov condition. We apply the approach to obtain new bounds on the expected running time of several algorithms, including Decentralized SGD (DSGD), a variant of decentralized SGD, known as *Stacked SGD*, and the stochastic variance reduced gradient (SVRG) algorithm. Finally, we consider the generalization properties of the iterate returned by early stopping.

## 1 INTRODUCTION

This work considers the minimization of a differentiable and possible nonconvex objective function:

$$\min_{x \in \mathbb{R}^d} f(x). \tag{1}$$

A generally accepted success criteria for algorithms that use only first-order information is that an *approximate stationary point* is generated. These are points $x \in \mathbb{R}^d$ at which the function $f$ has a small gradient. In a typical machine learning scenario, $f$ is the average loss over a dataset of training examples, and the method of choice involves using some form of stochastic gradient method, for instance, stochastic gradient descent, or SGD (see Algorithm 1). The success of SGD in machine learning problems has led to many extensions of the algorithm, including variance-reduced and distributed variants (reviewed in Section 1.1).

A common approach to stopping optimization in practice is to use early stopping based on a validation function. In this scenario, a stopping criterion is periodically evaluated on the validation function, and the algorithm stops once this criterion is met. The validation and training sets often are disjoint (although this is not required in the present work). Although this approach is used frequently, there is little theoretical work on the runtime of nonconvex optimization using early stopping based on a validation function. In general, the runtime and performance will depend on several factors, including the relation between the validation and training functions, and the desired level of solution accuracy. In this work, the stopping criterion is that the algorithm has generated a point, which is an approximate stationary point for the validation function. Our analysis focuses on bounding the number of iterations and gradient evaluations used until the algorithm meets the stopping criteria. Formally, we consider the stopping time defined as the first time an iterate has the property of being an approximate stationary point for the validation function, and we derive upper bounds on the expected value of this stopping time. Using bounds on the Wasserstein distance between the training and validation sets, we also may derive a bound on the stationary gap of the training function at the resulting iterate. As an extension, we also describe how Wasserstein concentration bounds can be used to bound the stationarity gap with respect to the testing distribution to which both the training and validation datasets are drawn.

We apply our analysis to several settings, including stochastic gradient descent (SGD), stochastic variance reduced gradient (SVRG), decentralized SGD (DSGD) and stacked SGD (SSGD). The

result is new bounds on the expected number of Incremental First-order Oracle (IFO) calls needed to generate approximate stationary points for some known algorithms (SGD; DSGD; SVRG), as well as a new algorithm (SSGD).

**Main contributions**   Our main contributions include:

– We present a non-asymptotic analysis of SGD with early stopping that leads to a bound on the expected number of gradient evaluations needed to find approximate stationary points of the training function (Corollary 3.5). The analysis allows for biases in the update direction, subject to a Lyapunov-type inequality on the error terms.

– We rigorously analyze the expected running time of two distributed SGD algorithms: Decentralized SGD, and Stacked SGD (Algorithm 2), a new decentralized form of SGD designed to exploit connectivity patterns consisting of a network of parameter-server type clusters (Corollary 4.4.)

– We apply the analysis to nonconvex SVRG to obtain a bound on the expected number of IFO calls for this algorithm (Corollary 6.2).

– We demonstrate how Wasserstein concentration bounds can be leveraged to bound the generalization performance of parameters returned by SGD with early stopping (Corollary 7.2).

## 1.1   RELATED WORK

The study of stochastic optimization goes back (at least) to the pioneering efforts of Robbins and Monro Robbins & Monro (1951). Subsequent developments include the ordinary differential equation (ODE) method Ljung (1977) and stochastic approximation Kushner & Clark (1978), which emphasizes the asymptotic behavior of the algorithms.

The subject of non-asymptotic performance guarantees has attracted interest as well, including lower bounds on algorithm performance Nemirovski & Yudin (1983). For a review of non-asymptotic guarantees for convex SGD, the reader may consult Nemirovski et al. (2009); Rakhlin et al. (2012); Bach & Moulines (2013). Extensions such as distributed Zinkevich et al. (2010) and asynchronous Zhang et al. (2015); Agarwal & Duchi (2011) variants of the algorithms also have been investigated. Many machine learning optimization problems involve an objective represented as a finite sum of functions, and, for this case, variance reduction techniques lead to improved rates of convergence over SGD Johnson & Zhang (2013); L. Roux et al. (2012); Defazio et al. (2014).

The randomized stochastic gradient (RSG) method Ghadimi & Lan (2013) uses randomization to obtain a non-asymptotic performance guarantee for SGD. The randomization technique has became a standard tool for analyzing optimization algorithms in the nonconvex setting Ghadimi & Lan (2016); Lian et al. (2017); Allen-Zhu (2018a;b); Reddi et al. (2016b;a); Zhang et al. (2016); Reddi et al. (2016b); Lei et al. (2017); Lian et al. (2015). Follow-up works have included analysis of nonconvex optimization in more sophisticated algorithmic settings, such as asynchronous Lian et al. (2015) and decentralized Lian et al. (2017) optimization. Analysis of variance reduced optimization has extended beyond convex functions beginning with an application to principal components analysis Shamir (2015) and later to general nonconvex functions Allen-Zhu & Hazan (2016); Reddi et al. (2016b;a); Lei et al. (2017). A highlight in this area is that the IFO complexity of SVRG for nonconvex functions is superior to that of RSG Allen-Zhu & Hazan (2016); Reddi et al. (2016a). Algorithms with better convergence rates than SVRG also have been developed Allen-Zhu (2018a;b).

We are particularly interested in results for biased SGD, e.g., Bertsekas & Tsitsiklis (2000) which considers the asymptotic convergence of biased SGD to stationary points. In this work, we are interested in a similar scenario but instead focus on the non-asymptotic behavior of the algorithm.

Several recent works also have explored the average amount of resources needed to reach a desired performance level in optimization. The expected running time of a stochastic trust region algorithm (STORM) is given in Blanchet et al. (2016). A similar methodology also has been used to analyze stochastic line search methods Paquette & Scheinberg (2018). Our convergence analysis is similar in spirit to these works as we also are interested in the expected amount of time or other resources required to meet the performance guarantee. However, the algorithms and assumptions in this work differ.

Other recent work has analyzed the theoretical aspects of early stopping. For instance, Duvenaud et al. (2016) developed an interpretation of early stopping in terms of variational Bayesian inference. Early stopping for a least squares problem in a reproducing kernel Hilbert space has been treated in Lin & Rosasco (2016). The implications of early stopping generalization were studied in Hardt et al. (2016). However, to our knowledge, this work is the first to analyze the algorithms' runtimes when using a validation function for early stopping in nonconvex optimization.

## 2 PRELIMINARIES

Let $f : \mathbb{R}^q \times \mathbb{R}^d \to \mathbb{R}$ be a loss function whose value we denote by $f(y, x)$. Intuitively, the variable $y$ represents an input/output pair, and $x$ represents the model parameters. Throughout, we shall assume the gradient of the objective function with respect to $x$ is Lipschitz continuous (defined as follows).

**Assumption 2.1.** *The function $f : \mathbb{R}^q \times \mathbb{R}^d \to \mathbb{R}$ is bounded from below by $f^* \in \mathbb{R}$, and the derivative of $f$ with respect to $x$ is $L$-Lipschitz continuous:*

$$\forall y \in \mathbb{R}^q, x_1, x_2 \in \mathbb{R}^d, \quad \|\nabla_x f(y, x_1) - \nabla_x f(y, x_2)\| \leq L\|x_1 - x_2\|.$$

At times, we will make a distinction between the training function $f_T$, which is used to calculate gradients, and a validation function $f_V$ used to decide when to stop training.

**Assumption 2.2.** *The training function $f_T$ is defined using a set $Y_T \subseteq \mathbb{R}^q$ of $n_T$ elements as $f_T(x) = \frac{1}{n_T} \sum_{y \in Y_T} f(y, x)$, and the validation function $f_V$ is defined using a set $Y_V \subseteq \mathbb{R}^q$ of $n_V$ elements as $f_V(x) = \frac{1}{n_V} \sum_{y \in Y_V} f(y, x)$.*

To guarantee that the early stopping rule leads to a well-defined algorithm, we will assume a relation between the training and validation functions. Intuitively, the functions $f_T$ and $f_V$ will be similar when the datasets $Y_T$ and $Y_V$ are similar. Formally, the datasets $Y_T$ and $Y_V$ determine probability measures $\mu_T$ and $\mu_Y$, defined as $\mu_T = \frac{1}{n_T} \sum_{y \in Y_T} \delta_y$ and $\mu_V = \frac{1}{n_V} \sum_{y \in Y_V} \delta_y$, respectively, where $\delta_y$ is the delta measure $\delta_y(A) = 1_{y \in A}$ for all sets $A$. We can compare these measures using the Wasserstein distance as in:

For $q \geq 1, p \geq 1$, we denote by $\mathcal{P}_p(\mathbb{R}^q)$ the probability measures on $\mathbb{R}^q$ with finite moments of order $p$. Recall that a coupling of probability measures $\mu_1$ and $\mu_2$ is a probability measure $\gamma$ on $\mathbb{R}^q \times \mathbb{R}^q$ such that for all measurable sets $A$, $\gamma(A \times \mathbb{R}^q) = \mu_1(A)$ and $\gamma(\mathbb{R}^q \times A) = \mu_2(A)$. Intuitively, a coupling transforms data distributed like $\mu_1$ into a dataset that is distributed according to $\mu_2$. The $p$-Wasserstein distance on $\mathcal{P}_p(\mathbb{R}^q)$, denoted by $d_p$, is defined as:

$$d_p(\mu_1, \mu_2) = \inf_{\gamma \in \Gamma(\mu_1, \mu_2)} \left( \mathbb{E}_{(x_1, x_2) \sim \gamma} \left[ \|x_1 - x_2\|^p \right] \right)^{1/p}, \tag{2}$$

where $\Gamma(\mu_1, \mu_2)$ is the set of all couplings of $\mu_1$ and $\mu_2$. For more details the reader is referred to Villani (2008).

**Assumption 2.3.** *There is a constant $G \geq 0$ such that*

$$\forall x \in \mathbb{R}^d, \quad \|\nabla f_V(x) - \nabla f_T(x)\| \leq G d_1(\mu_V, \mu_T).$$

There are several cases in which this assumption will be satisfied. It is trivially satisfied if the training and validation sets are the same as $\mu_T = \mu_V$. As a consequence of the Kantorovich duality formula (Villani (2008), Remark 6.5), it is also satisfied if the function $y \mapsto \nabla_x f(y, x)$ is a $G$-Lipschitz function, uniformly for all $x$. Consider the following example:

**Example 2.4.** Suppose that $g : \mathbb{R}^q \times \mathbb{R}^d \to \mathbb{R}$ is a smooth function. Let $h : \mathbb{R}^d \to \mathbb{R}$ be the function that applies the hyperbolic tangent function to each of its components: $h(x) = (\tanh(x_1), \dots, \tanh(x_d))$, and define $f(y, x) = g(y, h(x))$. Further suppose that the training data are bounded: $\|y\| \leq J$ for all $y \in Y_T$. Then Assumption 2.3 is satisfied because $y \mapsto \nabla_x f(x, y)$ is a Lipschitz function with $G = \sup_{\|y\| \leq J, \|x\| \leq \sqrt{d}} \|\frac{\partial^2 g}{\partial x \partial y}(y, x)\|$.

In our analyses the notion of success is that an algorithm generates an approximate stationary point:

**Definition 2.5.** A point $x \in \mathbb{R}^d$ is an $\epsilon$-*approximate stationary point* of $f$ if $\|\nabla f(x)\|^2 \leq \epsilon$.

We measure the complexity of algorithms according to how many function value and gradient queries they make. Formally, an IFO) is defined as follows (Agarwal & Bottou, 2015):

**Definition 2.6.** An IFO takes a parameter $x$ and an input $y$ and returns the pair $(f(y, x), \nabla_x f(y, x))$.

In Appendix A.1, we briefly recall the notion of filtration, stopping times, and other concepts from stochastic processes that will be used in our analyses. We also refer readers to Williams (1991) for more details.

## 3 BIASED SGD

This section details our analysis of SGD with early stopping shown on the right in Algorithm 1. Starting from an initial point $x_1$, at each iteration, the parameter is updated with an approximate gradient $h_n$ using a step-size $\eta$. The gradient norm of the validation function is evaluated every $m$ iterations, and the algorithm ends when the norm decreases below a threshold $\epsilon$.

---
**Algorithm 1** SGD with early stopping
---
1: **input:** Initial point $x_1 \in \mathbb{R}^d$
2: $t = 1$
3: **while** $\|\nabla f_V(x_t)\|^2 > \epsilon$ **do**
4:     **for** $n = t$ **to** $t + m - 1$ **do**
5:         $x_{n+1} = x_n - \eta h_n$
6:     **end**
7:     $t = t + m$
8: **end**
9: **return** $x_t$
---

We assume that the update direction $h_t$ is a sum of two components, $v_t$ and $\Delta_t$, that represent an unbiased gradient estimate and an error term, respectively:

$$h_t = v_t + \Delta_t. \tag{3}$$

Let $\{\mathcal{F}_t\}_{t \geq 0}$ be a filtration such that $x_1$ is $\mathcal{F}_0$-measurable, and for all $t > 1$, the variables $v_t, \Delta_t$ are $\mathcal{F}_t$-measurable. Our assumptions on the $v_t$ are as follows.

**Assumption 3.1.** *For any $t \geq 1$, it holds that*

$$\mathbb{E}\left[v_t - \nabla f_T(x_t) \mid \mathcal{F}_{t-1}\right] = 0, \tag{4}$$

$$\mathbb{E}\left[\|v_t - \nabla f_T(x_t)\|^2 \mid \mathcal{F}_{t-1}\right] \leq \sigma_v^2. \tag{5}$$

Assumption 3.1 states that the update directions $v_t$ are valid approximations to the gradient $\nabla f_T(x_t)$, and it also bounds the error in the approximation. For the variables $\Delta_t$ we assume the following:

**Assumption 3.2.** *There is a sequence of random variables $V_1, V_2, \ldots,$ and $U_1, U_2, \ldots$ such that for all $t \geq 1$ the variables $V_t$ and $U_t$ are $\mathcal{F}_t$-measurable, $\|\Delta_t\|^2 \leq V_t$, and the $V_t$ satisfy the following Lyapunov-type inequality: For constants $\alpha \in [0, 1)$ and $\beta \geq 0$,*

$$V_1 \leq \beta, \tag{6}$$

$$\forall t \geq 2, \quad V_t \leq \alpha V_{t-1} + U_{t-1}, \tag{7}$$

$$\forall t \geq 1, \quad \mathbb{E}\left[U_t \mid \mathcal{F}_{t-1}\right] \leq \beta. \tag{8}$$

Assumption 3.2 models a scenario where the gradient error dynamics is a combination of contracting and expanding behaviors. Contraction shrinks the error and is represented by a factor $\alpha$. External noise, represented by the $U_t$ terms, prevents the error from vanishing completely. Note that the assumption would be satisfied in the unbiased case by simply setting $V_t = 0$.

We can now state our result on the expected number of iterations for SGD with early stopping:

**Proposition 3.3.** *Let $\{x_t\}_{t \geq 1}$ be as in Algorithm 1. Let Assumptions 2.1, 2.2, 2.3, 3.1, and 3.2 hold. For $\epsilon > 0$, let $\tau(\epsilon)$ be the stopping time $\tau(\epsilon) = \inf\{n \geq 1 \mid n \equiv 1 \pmod{m} \text{ and } \|\nabla f_V(x_n)\|^2 \leq \epsilon\}$. Suppose that $\eta \leq \frac{1}{L}$ and $\epsilon - 2Lm\eta\sigma_v^2 - 2m\beta/(1-\alpha) - G^2 d_1(\mu_V, \mu_T)^2 > 0$. Then*

$$\mathbb{E}[\tau(\epsilon)] \leq \frac{\eta G^2 d_1(\mu_V, \mu_T)^2 + 2(f_T(x_1) - f^*) + \eta\epsilon/m + 2\eta\beta/(1-\alpha)}{\eta\epsilon/m - 2L\eta^2\sigma_v^2 - 2\eta\beta/(1-\alpha) - \eta G^2 d_1(\mu_V, \mu_T)^2/m}. \tag{9}$$

*Furthermore, it holds with probability 1 that $\|\nabla f_T(x_{\tau(\epsilon)})\|^2 \leq 2\epsilon + 2G^2 d_1(\mu_V, \mu_T)^2$.*

This result can be strengthened by assuming a coupling between the step-size and the expansion bound $\beta$, then the result can be strengthened as demonstrated in the next corollary.

**Corollary 3.4.** *Let Assumptions 2.1, 2.2, 2.3, 3.1, and 3.2 hold. In the context of Proposition 3.3, let the constant $\beta$ be of the form $\beta = \eta R$ for some $R \geq 0$, and suppose that $\epsilon > G^2 d_1(\mu_V, \mu_T)^2$. Let $c \in (0,1)$ and let the step-size be*

$$\eta = c \cdot \min\left\{\frac{1}{L}, \frac{\epsilon - G^2 d_1(\mu_V, \mu_T)^2}{m(2L\sigma_v^2 + 2R/(1-\alpha))}\right\}. \tag{10}$$

*Then*

$$\mathbb{E}[\tau(\epsilon)] = \mathcal{O}\left(\frac{m^2\left(1 + R/(1-\alpha)\right)}{(1-c)\,c\,(\epsilon - G^2 d_1(\mu_V, \mu_T)^2)^2}\right) \tag{11}$$

*See the appendix for the complete formula, including lower order terms.*

From this corollary we see that when $\beta$ is proportional to the step-size $\eta$, the complexity bound is of the order $\mathcal{O}(1/\epsilon^2)$. As we shall see below, this coupling assumption is satisfied for Stacked SGD.

For the case of using SGD to minimize a finite sum using unbiased, we obtain the following:

**Corollary 3.5.** *Let Assumptions 2.1, 2.2, and 2.3 hold. Suppose each gradient estimate is obtained by selecting a data point $y_t \in Y_T$ uniformly at random and setting $v_t = \nabla_x f(y_t, x_t)$ and that there is a $\sigma_v^2 \geq 0$ such that $\forall\, y' \in Y_T, x \in \mathbb{R}^d$, $\left\|\nabla_x f(y', x) - \frac{1}{n_T}\sum_{y \in Y_T} \nabla_x f(y, x)\right\|^2 \leq \sigma_v^2$. If the step-sizes are defined according to Equation equation 20 using $c = 1/2$ then the expected number of IFO calls used by SGD before reaching an $\epsilon$-approximate stationary point is*

$$\mathbb{E}\left[\text{IFO}(\epsilon)\right] = \mathcal{O}\left(\frac{mn_V + m^2}{(\epsilon - G^2 d_1(\mu_V, \mu_T)^2)^2} + n_V\right)$$

*See the appendix for the complete formula, including lower-order terms.*

Note that when $d(\mu_V, \mu_T)$ is on the order of $\sqrt{\epsilon}$, this result states that the expected IFO complexity is $\mathcal{O}(1/(\epsilon^2))$. This can be compared with the RSG algorithm, where $\mathcal{O}(1/(\epsilon^2))$ iterations are sufficient for the expected (squared) norm of the gradient at a random iterate to be at most $\epsilon$ (Cor. 2.2 in Ghadimi & Lan (2013)).

## 4 Stacked SGD

In this section, we introduce SSGD for decentralized optimization, which involves a distributed system with two types of nodes: workers and communicators. The communication pattern is shown on the right for a hypothetical system consisting of 16 workers and 4 communicators. Workers (circles) are grouped into clusters, each containing a communicator (triangles).

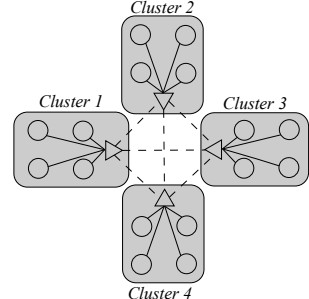

Algorithm 2 depicts the steps of SSGD. For every $m$ epochs, the gradient of the validation function is computed at the mean of the communication node parameters. The algorithm stops when the norm of this gradient drops below a threshold. This step is naturally carried out by one of the communication nodes because they can store the average computed during Line 9 of the algorithm. For workers, each iteration begins on Line 6 with one step of SGD. Then, on Line 7, an averaging step is performed to partially synchronize the model parameters within the local cluster. The steps for a communication node begin in Line 9 with an averaging step, among the other communication nodes from each cluster. On Line 10, there is a partial synchronization with the worker nodes in the cluster.

The key to the algorithm's efficiency is that the averaging among communication nodes can happen in parallel with the gradient descent step as it occurs within the cluster. In the naive approach to parallelizing SGD, all nodes must block after each iteration for synchronization of their parameters.

We assume there are $M \geq 1$ clusters, each containing $K \geq 1$ computation nodes and 1 communication node. Thus, there are $M(K + 1)$ total nodes. Given a worker node $1 \leq i \leq KM$, we let

---

**Algorithm 2** SSGD with early stopping

---

1: **input:** Node id $i$, initial parameters
   $x_1^i$ (for workers), or $\widehat{x}_1^i$ (for communicators)
2: $t = 1$
3: **while** $\left\| \nabla f_V \left( \frac{1}{M} \sum_{j=1}^M \widehat{x}_t^j \right) \right\|^2 > \epsilon$ **do**
4:     **for** $n = t$ **to** $t + m - 1$ **do**
5:         **if** node $i$ is a computation node **then**
6:             $x_{n+\frac{1}{2}}^i = x_n^i - \eta v_n^i$
7:             $x_{n+1}^i = \frac{1}{K+1} \left( \widehat{x}_{n+\frac{1}{2}}^{c(i)} + \sum_{j=1}^K x_{n+\frac{1}{2}}^j \right)$
8:         **else** (node $i$ is a communication node)
9:             $\widehat{x}_{n+\frac{1}{2}}^i = \frac{1}{M} \sum_{j=1}^M \widehat{x}_n^j$
10:            $\widehat{x}_{n+1}^i = \frac{1}{K+1} \left( \widehat{x}_{n+\frac{1}{2}}^i + \sum_{j \in c^{-1}(i)} x_{n+\frac{1}{2}}^j \right)$
11:        **end if**
12:    **end**
13:    $t = t + m$
14: **end**
15: **return** $\widehat{x}_{t+\frac{1}{2}}$

---

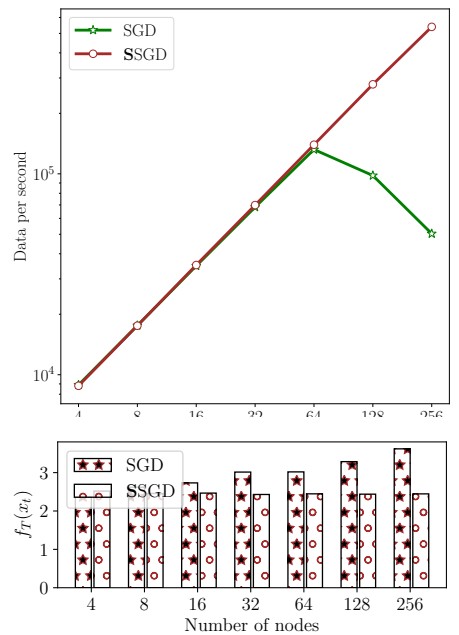

SSGD results for a climate modeling task. Top: throughput vs number of nodes.. Bottom: comparison of the loss of the algorithms.

$c(i) \in \{1, \dots, M\}$ denote the index of the communication node of the group containing worker $i$. For analysis, we define the filtration $\{\mathcal{F}_t\}_{t \geq 0}$ as follows:

$$\mathcal{F}_0 = \sigma\big( \{ x_1^i \,|\, 1 \leq i \leq KM \} \cup \{ \widehat{x}_1^i \,|\, 1 \leq i \leq M \} \big),$$

$$\forall t \geq 1, \quad \mathcal{F}_t = \sigma\big( \{ x_1^i, v_n^i \,|\, 1 \leq n \leq t, \, 1 \leq i \leq KM \} \cup \{ \widehat{x}_1^i \,|\, 1 \leq i \leq M \} \big).$$

We assume that the gradient estimates used in SSGD are unbiased and have bounded variance:

**Assumption 4.1.** *For any $t \geq 1$ and $1 \leq i \leq KM$,*

$$\mathbb{E}\left[ v_t^i - \nabla f_T(x_t^i) \mid \mathcal{F}_{t-1} \right] = 0, \tag{12}$$

$$\mathbb{E}\left[ \left\| v_t^i - \nabla f_T(x_t^i) \right\|^2 \mid \mathcal{F}_{t-1} \right] \leq \sigma_v^2. \tag{13}$$

The first step in our analysis is a bound on the dispersion of the parameters across the system.

**Proposition 4.2.** *Let Assumption 2.1, 2.2, and 4.1 hold, and let the variables $\widehat{x}_t^i$ be as defined in Line 6 of Algorithm 2. Suppose the step-size satisfies $\eta < 1/(2LK)$. Define the variables $V_1, U_1, V_2, U_2, \dots$ and the constants $\alpha, \beta$ as follows:*

$$V_t = \frac{L^2}{M^2} \sum_{i=1}^M \sum_{j=1}^M \left\| \widehat{x}_t^i - \widehat{x}_t^j \right\|^2, \tag{14a}$$

$$U_t = \frac{\eta K L}{(K + 5/8)} \frac{1}{M^2} \sum_{i=1}^M \sum_{j=1}^M \left\| \left( \frac{1}{K} \sum_{k \in c^{-1}(i)} v_t^k - \nabla f_T(\widehat{x}_t^i) \right) - \left( \frac{1}{K} \sum_{k \in c^{-1}(j)} v_t^k - \nabla f_T(\widehat{x}_t^j) \right) \right\|^2 \tag{14b}$$

$$\alpha = \frac{(K + 3/4)^2}{(K+1)^2}, \tag{14c}$$

$$\beta = \eta \cdot \frac{4 L \sigma_v^2}{(K + 5/8)}. \tag{14d}$$

*Then, for all $t \geq 1$, it holds that $V_{t+1} \leq \alpha V_t + U_t$ and $\mathbb{E}[U_t \mid \mathcal{F}_{t-1}] \leq \beta$.*

In the preceding definitions, $V_t$ represents the dispersion of the parameter values across different nodes. An averaging step tends to reduce the dispersion by a factor of $\alpha$, while the independent gradient updates at each node may increase parameter dispersion by an amount $\beta$. This result allows us to model SSGD as a form of biased SGD, leading to the following:

**Proposition 4.3.** *Let Assumptions 2.1, 2.2, 2.3, and 4.1 hold. Assume that the initial parameters at every node are equal:* $x_1^i = x_1^j$, *for all* $1 \leq i, j \leq KM$ *and* $\widehat{x}_1^i = \widehat{x}_1^j$ *for* $1 \leq i, j \leq M$. *For some* $c < \frac{1}{2K}$, *suppose that the step-size* $\eta$ *is*

$$\eta = c \cdot \min \left\{ \frac{1}{L}, \frac{\epsilon - G^2 d_1(\mu_V, \mu_T)^2}{m(2L\sigma_v^2/K + 32L\sigma_v^2(K+1)/(K+5/8))} \right\}. \tag{15}$$

*Let* $\widehat{x}_t$ *be the average of the communicator states at time* $t$: $\widehat{x}_t = \frac{1}{M} \sum_{i=1}^{M} \widehat{x}_t^i$. *For* $\epsilon > 0$, *define* $\tau(\epsilon)$ *to be the stopping time* $\tau(\epsilon) = \inf\{n \geq 1 \mid n \equiv 1 \pmod{m} \text{ and } \|\nabla f_V(\widehat{x}_n)\|^2 \leq \epsilon\}$. *Then,*

$$\mathbb{E}[\tau(\epsilon)] = \mathcal{O}\left( \frac{m^2}{(1-c)c(\epsilon - G^2 d_1(\mu_V, \mu_T)^2)^2} \right).$$

*Refer to the Appendix for the complete formula, including lower-order terms.*

This leads to a bound on the expected IFO complexity to minimize a finite sum using SSGD:

**Corollary 4.4.** *Let Assumptions 2.1, 2.2, and 2.3 hold. Suppose each gradient estimate is obtained by selecting a data point* $y_t^j \in Y_T$ *uniformly at random and setting* $v_t^j = \nabla_x f(y_t^j, x_t^j)$ *and that there is a* $\sigma^2 \geq 0$, *such that* $\forall y' \in Y_T$, $x \in \mathbb{R}^d$, $\left\| \nabla_x f(y', x) - \frac{1}{n_T} \sum_{y \in Y_T} \nabla_x f(y, x) \right\|^2 \leq \sigma_v^2$. *If the step-sizes are defined as in Equation equation 15 with* $c = 1/(4K)$, *then the expected number of IFO calls used by SSGD before reaching an* $\epsilon$-*approximate stationary point is*

$$\mathbb{E}[\text{IFO}(\epsilon)] = \mathcal{O}\left( \frac{mK(n_V + mK)}{(\epsilon - G^2 d_1(\mu_V, \mu_T)^2)^2} + n_V \right).$$

**Experimental Result** SSGD has been implemented to train a neural network model as part of research into spatio-temporal data analysis for climate research. The model is LSTNet, a neural network architecture that includes a convolutional neural network to extract short-term local dependency patterns from spatial variables and a recurrent neural network component to discover long-term patterns from time series trends Lai et al. (2018). LSTNet is trained for the prediction of solar radiation from past sensor measurements. Fig. 4 shows results from the experiment. The upper plot compares the throughput of SGD and SSGD as the number of nodes increases. We see that the performance of SGD degrades after 64 nodes, while SSGD maintains near-linear scalability. The lower plot shows that although SSGD involves biased gradients, it does not sacrifice accuracy, and yields similar prediction errors compared to SGD. See Appendix G.1 for more details about the experimental methodology.

## 5 DECENTRALIZED SGD

In this section we show how our methodology can be applied to Decentralized SGD (DSGD) which is another variant of distributed SGD. Recently, DSGD was analyzed using randomization Lian et al. (2017). In this section we complement that analysis by studying the expected running time of the algorithm.

The steps of DSGD are shown in Algorithm 3. The procedure involves $M > 0$ worker nodes that participate in the optimization. A communication matrix $a$ describes the connectivity among the workers; $a_{i,j} > 0$ means that workers $i$ and $j$ will communicate after each gradient descent step. At each step of optimization, every node computes a weighted average of the parameters in its local neighborhood, as determined by the connectivity matrix. This is combined with a local gradient approximation to obtain the new parameter at the worker. The data that is returned by the algorithm (assuming the termination criteria is met) is the average of the parameters throughout the system, denoted $\overline{x}_t$:

$$\overline{x}_t = \frac{1}{M} \sum_{i=1}^{M} x_i \tag{16}$$

Every $m$ epochs, the norm of gradient of the validation function is evaluated at the average parameter and the algorithm terminates when this norm falls below a threshold.

The intuitive justification for DSGD is that it may be more efficient compared to naive approaches to parallelizing SGD, since whenever $a_{i,j} = 0$ then the nodes $i$ and $j$ need not communicate. In Lian et al. (2017) those authors offer theoretical support for the superiority of DSGD. In the present work, our goal is to analyze the expected running time of DSGD as an example of our the abstract theory developed above may be applied in practice. We leave comparisons of the algorithms for future work.

For the analysis, we define the filtration $\{\mathcal{F}_t\}_{t \geq 0}$ as follows:

$$\mathcal{F}_0 = \sigma\big(\{x_1^i \mid 1 \leq i \leq M\}\big),$$
$$\forall t \geq 1, \quad \mathcal{F}_t = \sigma\big(\{x_1^i, v_n^i \mid 1 \leq n \leq t,\, 1 \leq i \leq M\}\big)$$

---

**Algorithm 3** DSGD with early stopping

---
1: **input:** Initial point $x_1 \in \mathbb{R}^d$, initial parameters $x_1^i$.
2: $t = 1$
3: **while** $\|\nabla f_V(\overline{x}_t)\|^2 > \epsilon$ **do**
4:    **for** $n = t$ **to** $t + m - 1$ **do**
5:       $x_{n+1}^i = \sum\limits_{j=1}^{M} a_{i,j} x_n^j - \eta v_n^i$
6:    **end**
7:    $t = t + m$
8: **end**
9: **return** $\overline{x}_t$

---

The connectivity matrix $a$ is subject to the same conditions as in Lian et al. (2017):

**Assumption 5.1.** *The $M \times M$ connectivity matrix $a$ is symmetric and stochastic. The spectral gap, denoted by $\rho$ and defined as $\rho = (\max\{|\lambda_2(a)|, |\lambda_M(a)|\})^2$ is assumed to satisfy $\rho < 1$.*

To make the proofs clear and concise, we make the assumption the parameters at each node are single real-numbers. That is, throughout this section we assume $d = 1$ in Assumption **??**.

We also assume that the gradient estimates used at each worker at unbiased and have bounded variance.

**Assumption 5.2.** *For any $t \geq 1$ and $1 \leq i \leq M$,*

$$\mathbb{E}\left[v_t^i - \nabla f_T(x_t^i) \mid \mathcal{F}_{t-1}\right] = 0, \tag{17}$$
$$\mathbb{E}\left[|v_t^i - \nabla f_T(x_t^i)|^2 \mid \mathcal{F}_{t-1}\right] \leq \sigma_v^2. \tag{18}$$

For the analysis, we show that the sequence of averages $\overline{x}_t$ for $t = 1, 2, \ldots$ can be modeled as being generated by a biased version of SGD, using the tools from Section 3. This involves showing that the distance between local parameter values $x_t^i$ and the system average can be controlled, as shown in the following.

**Proposition 5.3.** *Let Assumptions 2.1, 2.2, 5.1, and 5.2 hold. Suppose the step-size satisfies $\eta \leq (1 - \sqrt{\rho})/(4\sqrt{2}L)$. Define the variables $V_1, U_1, V_2, U_2, \ldots$ and the constants $\alpha, \beta$ as follows:*

$$V_t = \frac{L^2}{M} \sum_{i=1}^{M} |x_t^i - \overline{x}_t|^2, \tag{19a}$$

$$U_t = 8\,\eta^2 \frac{L^2(1 + \rho)}{M(1 - \rho)} \sum_{i=1}^{M} |v_n^i - \nabla f(x_t^i)|^2, \tag{19b}$$

$$\alpha = \frac{(1 + \rho)}{2\rho} \left(\frac{\sqrt{\rho} + 1}{2}\right)^2 \tag{19c}$$

$$\beta = \eta \frac{L\sqrt{2}}{1 - \rho} \sigma_v^2 \tag{19d}$$

*Then for all $t \geq 1$ it holds that $V_{t+1} \leq \alpha V_t + U_t$ and $\mathbb{E}[U_t \mid \mathcal{F}_{t-1}] \leq \beta$.*

Using this result on the dispersion of the parameters, we can move to the main result on decentralized SGD. The result gives conditions that guarantee the expected time $\mathbb{E}[\tau(\epsilon)]$ is finite, and also bounds this time in terms of the problem data. Notably, it shows a dependence on $\rho$, which is the mixing rate of the connectivity matrix.

**Proposition 5.4.** *Let Assumptions 2.1, 2.2, 5.1, 2.3 and 5.2 hold. Assume that the initial parameters at every node are equal: $x_1^i = x_1^j$ for all $1 \leq i, j \leq M$. Let $c \leq \frac{1 - \sqrt{\rho}}{4\sqrt{2}}$ and define $R = L\sqrt{2}\sigma_v^2/(1 - \rho)$*

**Algorithm 4** SVRG with early stopping

1: **input:** Initial point $x_m^1 \in \mathbb{R}^d$
2: **for** $s = 1, 2, \ldots$ **do**
3:      $x_0^{s+1} = x_m^s$
4:      $g^{s+1} = \frac{1}{n_T} \sum_{y \in Y_T} \nabla f_T(y, x_0^{s+1})$
5:      **if** $\|g^{s+1}\|^2 \leq \epsilon$ **then return** $x_0^{s+1}$
6:      **for** $t = 0$ **to** $m - 1$ **do**
7:          Sample $y_t^s$ uniformly at random from $Y_T$
8:          $v_t^s = \nabla f(y_t^s, x_t^{s+1}) - \nabla f(y_t^s, x_0^{s+1}) + g^{s+1}$
9:          $x_{t+1}^{s+1} = x_t^{s+1} - \eta v_t^s$
10:      **end**
11: **end**

IFO calls to approximate stationarity for MNIST (top) and CIFAR-10 (bot.).

*and let $\alpha$ be as in Equation equation 19c. Let the step-size be*

$$\eta = c \cdot \min \left\{ \frac{1}{L}, \frac{\epsilon - G^2 d_1(\mu_V, \mu_T)^2}{m(2L\sigma_v^2 + 2R/(1-\alpha))} \right\}. \tag{20}$$

*For $\epsilon > 0$ define $\tau(\epsilon)$ to be the first time the norm of the gradient of the validation function falls below $\epsilon$; That is, $\tau(\epsilon) = \inf\{n \geq 1 \mid n \equiv 1 \ (\mathrm{mod}\ m) \ \text{and} \ \|\nabla f_V(\overline{x}_n)\|^2 \leq \epsilon\}$. Then*

$$\mathbb{E}[\tau(\epsilon)] = \mathcal{O}\left( \frac{m^2 (1 + R/(1-\alpha))}{(1-c)\,c\,(\epsilon - G^2 d_1(\mu_V, \mu_T)^2)^2} \right) \tag{21}$$

Note that in the above result, the order of the convergence is the same as for regular SGD and stacked SGD. An interesting avenue for future work would be to explore whether it is possible to obtain bounds where the step-size condition does not depend on the epoch length $m$.

## 6 SVRG

We demonstrate how the method can applied to a variant of the SVRG Johnson & Zhang (2013) with early stopping, shown in Algorithm 4. Each epoch begins with a full gradient computation (Line 4), and then an inner loop runs for $m$ steps. The first step of the inner loop is to choose a random data point (Line 7). Then, the update direction is computed (Line 8) and used obtain the next parameter (Line 9).

Combining some existing bounds for SVRG with our stopping time approach yields the following bound on the expected number of iterations until SVRG with early stopping terminates:

**Proposition 6.1.** *Let Assumptions 2.1 and 2.2 hold and consider the variables $x_t^{s+1}$ defined by Algorithm 4. Let $\xi = 1/4$ and suppose that the step-size is set to $\eta = \xi/(Ln_T^{2/3})$ and the epoch length is $m = \lfloor n_T/(3\xi) \rfloor$. For $\epsilon > 0$, define $\tau(\epsilon)$ to be the stopping time $\tau(\epsilon) = \inf\left\{ s \geq 1 \mid \|\nabla f_T(x_0^{s+1})\|^2 \leq \epsilon \right\}$. Then, $\mathbb{E}[\tau(\epsilon)] \leq 1 + (40Ln_T^{2/3}(f_T(x_m^1) - f^*))/\epsilon$.*

Note that Proposition 6.1 counts the number of epochs until an approximate stationary point is generated. A bound on the number of IFO calls can be obtained by multiplying $\tau$ by the number of IFO calls per epoch, which is $n_T + 2m$. This immediately leads to the following result:

**Corollary 6.2.** *Let Assumptions 2.1 and 2.2 hold and suppose the step-size $\eta$ and epoch length $m$ are defined as in Proposition 6.1. Then, the expected number of IFO calls until SVRG returns an approximate stationary point is $\mathbb{E}\left[\mathrm{IFO}\,(\epsilon)\right] = \mathcal{O}((n_T^{5/3}/\epsilon) + n_T)$.*

This result may be compared with Cor. 4 of Reddi et al. (2016a), which concerns an upper bound on the IFO calls for the expected (squared) norm of the gradient at a randomly selected iterate to be less than $\epsilon$. Our result concerns the expected number of IFO calls before the algorithm terminates with an iterate that is guaranteed to be an approximate stationary point with probability 1, a stronger property.

Fig. 6 illustrates the expected IFO complexity of SVRG and SGD. The top plot shows the results of an experiment using a simple logistic classifier on the MNIST dataset, and the bottom shows the result of training a one-layer neural net on the CIFAR-10 dataset. The error bands represent the standard deviation of the measurements over five independent runs. Appendix G.2 includes more details about the experimental methodology. In each case, SVRG is better at obtaining accurate solutions, while, for SGD, the expected IFO calls seem to become unbounded for sufficiently small $\epsilon$.

## 7 GENERALIZATION PROPERTIES

Typically, the training and validation sets are made from independent samples of a test distribution $\mu$, and it is of interest to estimate the model performance on samples from this test distribution. Define $f_\mu : \mathbb{R}^d \to \mathbb{R}$ as $f(x) = \mathbb{E}_\mu[f(y,x)]$. In this section, we derive a bound on $\mathbb{E}[\|\nabla f_\mu(x_{\tau(\epsilon)})\|^2]$, where the expectation is not only over the variates generated by optimization, but also over the random choice of the datasets $Y_V$ and $Y_T$. We will show how Wasserstein concentration bounds, which concern the average distance between $\mu$ and its empirical versions, can be used for this task. Consider the following from Dereich et al. (2013).

**Theorem 7.1** (Dereich et al. (2013), Special case of Theorem 1). *Let $d \geq 3$ and let $\mu$ be a measure on $\mathbb{R}^d$, such that $J = \mathbb{E}_\mu\left[\|y\|^3\right]^{1/3} < \infty$. Then, there is a constant $\kappa_d$, such that $\mathbb{E}[d_2(\mu, \mu_V)^2] \leq \kappa_d J n_V^{-3/d}$.*

This leads to a bound on the generalization performance of the iterates returned by SGD with early stopping, whose proof is deferred to the Appendix.

**Corollary 7.2.** *Let the conditions of Proposition 3.3 hold. Further assume that $J = \mathbb{E}_\mu\left[\|y\|^3\right]^{1/3} < \infty$, the validation set $Y_V$ is an empirical version of $\mu$, and $y \mapsto \nabla_x f(y,x)$ is uniformly $G$-Lipschitz. If $x_\tau(\epsilon)$ is the output of Algorithm 1, then $\mathbb{E}[\|\nabla f_\mu(x_{\tau(\epsilon)})\|^2] \leq 2\epsilon + 2G^2 \kappa_d J n_V^{-3/d}$.*

Note that for strongly convex optimization, it is possible to obtain rates of convergence of the test error that are independent of the dimension Hsu & Sabato (2016). Corollary 7.2 is interesting as it accounts for data distribution properties (via the $3^{rd}$ moment $J$) and does not depend on the number of iterations used in SGD. This result could be compared with Hardt et al. (2016), where the authors proved a bound on the generalization gap for function values in terms of the number of iterations $T$ and samples in the training set $n_T$. There, the bound is independent of $d$ but increasing with $T$, while our bound is independent of the number of iterations in SGD. It will be interesting to determine if these two analyses can be combined.

## 8 DISCUSSION

This work presents an analysis of several stochastic gradient methods that use early stopping based on a validation function. We demonstrated that by blending existing analysis techniques with some basic tools related to stopping times, it is possible to bound the expected number of iterations and gradient evaluations to generate approximate stationary points. We also considered decentralized optimization and introduced a new algorithm, SSGD, that proved amenable to analysis in our framework. For SSGD, we obtained a convergence rate using our results for biased SGD, and experiments showed that the algorithm has favorable scaling properties compared to basic parallel SGD. Our application to SVRG demonstrated that the theoretical approach can be applied in various settings. We also considered the generalization properties of the output of early stopping. We hope these efforts inspire other works that investigate the theoretical and practical aspects of early stopping.

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

APPENDIX: ON THE EXPECTED RUNNING TIME OF NONCONVEX OPTIMIZATION
WITH EARLY STOPPING

## A PRELIMINARIES

Our analyses make use of a quadratic bound for the training function which follows from Assumption 2.1:

$$\forall x, v \in \mathbb{R}^n, \quad f_T(x + v) \leq f_T(x) + \nabla f_T(x)^T v + \frac{L}{2} \|v\|^2. \tag{22}$$

### A.1 STOCHASTIC PROCESSES

The formal setting of a stochastic optimization algorithm involves a probability space $(\Omega, \mathcal{F}, \mathbb{P})$, consisting of a sample space $\Omega$, a $\sigma$-algebra $\mathcal{F}$ of subsets of $\Omega$ and a probability measure $\mathbb{P}$ on the subsets of $\Omega$ that are in $\mathcal{F}$. The algorithm takes an initial point $x_1$ and defines a sequence of random variables $\{x_t(\omega)\}_{t>1}$. Intuitively $\Omega$ represents the random data used by the algorithm, such as indices used to define mini-batches. For ease of notation we will omit the dependence of random variates in the algorithms on $\omega \in \Omega$. A filtration $\{\mathcal{F}_t\}_{t=0,1,\ldots}$ is an increasing sequence of $\sigma$-algebras, with the interpretation that $\mathcal{F}_t$ represents the information available to an algorithm up to and including time $t$. A random variable $x : \Omega \to \mathbb{R}^d$ is said to be $\mathcal{F}_t$ measurable if it can be expressed in terms of the state of the algorithm up and including time $t$. A rule for stopping an algorithm is represented as a stopping time, which is a random variable $\tau : \Omega \to \{0, 1, \ldots, \infty\}$ with the property that the decision of whether to stop or continue at time $n$ is made based on the information up to and including time $n$.

The following proposition will be used through out our analysis of the different algorithms.

**Proposition A.1.** *Let $\tau$ be a stopping time with respect to a filtration $\{\mathcal{F}_t\}_{t=0,1,\ldots}$. Suppose there is a number $c < \infty$ such that $\tau \leq c$ with probability one. Let $x_1, x_2, \ldots$ be any sequence of random variables such that each $x_t$ is $\mathcal{F}_t$-measurable and $\mathbb{E}[\|x_t\|] < \infty$. Then*

$$\mathbb{E}\left[\sum_{t=1}^{\tau} x_t\right] = \mathbb{E}\left[\sum_{t=1}^{\tau} \mathbb{E}\left[x_t \mid \mathcal{F}_{t-1}\right]\right]. \tag{23}$$

*Proof.* This is a consequence of the optional stopping theorem (Theorem 10.10 in Williams (1991)). Define $S_0 = 0$ and for $t \geq 1$, let $S_t = \sum_{i=1}^{t} (x_i - \mathbb{E}[x_i \mid \mathcal{F}_{i-1}])$. Then $S_0, S_1, \ldots$ is a martingale with respect to the filtration $\{\mathcal{F}_t\}_{t=0,1,\ldots}$, and the optional stopping theorem implies $\mathbb{E}[S_\tau] = \mathbb{E}[S_0]$. But $\mathbb{E}[S_0] = 0$, and therefore $\mathbb{E}[S_\tau] = 0$, which is equivalent to Equation equation 23. $\square$

## B ANALYSIS OF BIASED SGD

PROOF OF PROPOSITION 3.3

*Proof.* For convenience, define the random variables $\delta_t$ for $t = 1, 2, \ldots$ as $\delta_t = v_t - \nabla f_T(x_t)$. From equation 22, it holds that

$$f_T(x_{t+1}) \leq f_T(x_t) - \eta \nabla f_T(x_t)^T \left(\nabla f_T(x_t) + \delta_t + \Delta_t\right) + \frac{L}{2} \eta^2 \|\nabla f_T(x_t) + \delta_t + \Delta_t\|^2.$$

Summing this over $t = 1, \ldots, m$ yields

$$
\begin{aligned}
f_T(x_{m+1}) \leq{}& f_T(x_1) - \sum_{t=1}^{m} \eta \nabla f_T(x_t)^T \left( \nabla f_T(x_t) + \delta_t + \Delta_t \right) \\
&+ \sum_{t=1}^{m} \frac{L}{2} \eta^2 \| \nabla f_T(x_t) + \delta_t + \Delta_t \|^2 \\
={}& f_T(x_1) - \sum_{t=1}^{m} \eta \left( 1 - \frac{L}{2} \eta \right) \| \nabla f_T(x_t) \|^2 - \sum_{t=1}^{m} \eta (1 - L\eta) \nabla f_T(x_t)^T \delta_t \qquad (24) \\
&+ \sum_{t=1}^{m} \frac{L}{2} \eta^2 \| \delta_t \|^2 - \sum_{t=1}^{m} \eta (1 - L\eta) \nabla f_T(x_t)^T \Delta_t + \sum_{t=1}^{m} \frac{L}{2} \eta^2 \| \Delta_t \|^2 \\
&+ \sum_{t=1}^{m} L\eta^2 \delta_t^T \Delta_t .
\end{aligned}
$$

Note that, in general, for any numbers $a, b$ it is the case that $|ab| \leq \frac{1}{2} a^2 + \frac{1}{2} b^2$. Then

$$
|\delta_t^T \Delta_t| \leq \| \delta_t \| \| \Delta_t \| \leq \frac{1}{2} \| \delta_t \|^2 + \frac{1}{2} \| \Delta_t \|^2 \qquad (25)
$$

and

$$
|\nabla f_T(x_t)^T \Delta_t| \leq \| \nabla f_T(x_t) \| \| \Delta_t \| \leq \frac{1}{2} \| \nabla f_T(x_t) \|^2 + \frac{1}{2} \| \Delta_t \|^2 . \qquad (26)
$$

Combining Equations equation 24, equation 25, and equation 26, we obtain

$$
\begin{aligned}
f_T(x_{m+1}) \leq{}& f(x_1) - \sum_{t=1}^{m} \eta \left( 1 - \frac{L}{2} \eta - \frac{1}{2}(1 - L\eta) \right) \| \nabla f_T(x_t) \|^2 \\
&- \sum_{t=1}^{m} \eta (1 - L\eta) \nabla f_T(x_t)^T \delta_t + \sum_{t=1}^{N} \left( \frac{L}{2} \eta^2 + \frac{1}{2} L\eta^2 \right) \| \delta_t \|^2 \\
&+ \sum_{t=1}^{N} \left( \frac{L}{2} \eta^2 + \frac{1}{2} \eta (1 - L\eta) + L\eta^2 \frac{1}{2} \right) \| \Delta_t \|^2 \\
={}& f(x_1) - \sum_{t=1}^{m} \eta \frac{1}{2} \| \nabla f_T(x_t) \|^2 - \sum_{t=1}^{m} \eta (1 - L\eta) \nabla f_T(x_t)^T \delta_t + \sum_{t=1}^{m} L\eta^2 \| \delta_t \|^2 \\
&+ \sum_{t=1}^{m} \frac{1}{2} \eta (1 + L\eta) \| \Delta_t \|^2 .
\end{aligned}
$$

Rearranging terms and noting that $f_T(x_{m+1}) \geq f^*$, this yields

$$
\begin{aligned}
\sum_{t=1}^{m} \eta \frac{1}{2} \| \nabla f_T(x_t) \|^2 \leq{}& f_T(x_1) - f^* - \sum_{t=1}^{m} \eta (1 - L\eta) \nabla f_T(x_t)^T \delta_t + \sum_{t=1}^{m} L\eta^2 \| \delta_t \|^2 \\
&+ \sum_{t=1}^{m} \frac{1}{2} \eta (1 + L\eta) \| \Delta_t \|^2 \\
&\qquad\qquad\qquad\qquad\qquad\qquad\qquad\qquad\qquad\qquad (27) \\
\leq{}& f(x_1) - f^* - \sum_{t=1}^{m} \eta (1 - L\eta) \nabla f_T(x_t)^T \delta_t + \sum_{t=1}^{m} L\eta^2 \| \delta_t \|^2 \\
&+ \sum_{t=1}^{m} \eta V_t .
\end{aligned}
$$

where in the second inequality we used the assumptions that $\eta \leq 1/L$ and $\| \Delta_t \|^2 \leq V_t$. Based on our assumption that $\| \nabla f_V(x) - \nabla f_T(x) \| \leq G d_1(\mu_V, \mu_T)$, it follows that

$$
\| \nabla f_V(x) \|^2 \leq 2 G^2 d_1(\mu_V, \mu_T)^2 + 2 \| \nabla f_T(x) \|^2 . \qquad (28)
$$

Also note that

$$\sum_{t=1}^{n} 1_{t \equiv 1 \ (\mathrm{mod}\ m)} = \left\lceil \frac{n}{m} \right\rceil \leq \frac{n}{m} + 1. \tag{29}$$

Combining equation 28 and equation 29 results in

$$\sum_{t=1}^{\tau(\epsilon) \wedge n} 1_{t \equiv 1 \ (\mathrm{mod}\ m)} \|\nabla f_V(x_t)\|^2 \leq \sum_{t=1}^{\tau(\epsilon) \wedge n} 1_{t \equiv 1 \ (\mathrm{mod}\ m)} G^2 d_1(\mu_V, \mu_T)^2$$

$$+ \sum_{t=1}^{\tau(\epsilon) \wedge n} 1_{t \equiv 1 \ (\mathrm{mod}\ m)} \|\nabla f_T(x_t)\|^2 \tag{30}$$

$$\leq G^2 d_1(\mu_V, \mu_T)^2 \left( \frac{(\tau(\epsilon) \wedge n)}{m} + 1 \right) + \sum_{t=1}^{\tau(\epsilon) \wedge n} \|\nabla f_T(x_t)\|^2.$$

For each $n \geq 1$ define $\tau(\epsilon) \wedge n$ to be the stopping time which is the minimum of $\tau(\epsilon)$ and the constant $n$. Applying Proposition A.1 and Assumption 4, it holds that

$$\mathbb{E}\left[ \sum_{t=1}^{\tau(\epsilon) \wedge n} \nabla f_T(x_t)^T \delta_t \right] = 0 \tag{31}$$

and using Proposition A.1 with Assumption 5 gives

$$\mathbb{E}\left[ \sum_{t=1}^{\tau(\epsilon) \wedge n} \|\delta_t\|^2 \right] \leq \sigma_v^2 \mathbb{E}[\tau(\epsilon) \wedge n]. \tag{32}$$

Next, according to conditions equation 6, and equation 7, it holds for any $m \geq 1$ and with probability one that

$$\sum_{t=1}^{m} V_t \leq \alpha \sum_{t=1}^{m} V_t + \sum_{t=1}^{m} U_t + \beta \tag{33}$$

and by equation 8 together with Proposition A.1,

$$\mathbb{E}\left[ \sum_{t=1}^{\tau(\epsilon) \wedge n} U_t \right] \leq \mathbb{E}[\tau(\epsilon) \wedge n]\beta. \tag{34}$$

Combining equation 33 and equation 34, then

$$\mathbb{E}\left[ \sum_{t=1}^{\tau(\epsilon) \wedge n} V_t \right] \leq \alpha \mathbb{E}\left[ \sum_{t=1}^{\tau(\epsilon) \wedge n} V_t \right] + (\mathbb{E}[\tau(\epsilon \wedge n)] + 1)\beta$$

which, upon rearranging, results in

$$\mathbb{E}\left[ \sum_{t=1}^{\tau(\epsilon) \wedge n} V_t \right] \leq (\mathbb{E}[\tau(\epsilon) \wedge n] + 1) \frac{\beta}{1 - \alpha}. \tag{35}$$

Furthermore, by definition of $\tau$,

$$\mathbb{E}\left[ \sum_{t=1}^{\tau(\epsilon) \wedge n} 1_{t \equiv 1 \ (\mathrm{mod}\ m)} \|\nabla f_V(x_t)\|^2 \right] \geq \mathbb{E}\left[ \sum_{t=1}^{(\tau(\epsilon) \wedge n)-1} 1_{t \equiv 1 \ (\mathrm{mod}\ m)} \|\nabla f_V(x_t)\|^2 \right] \tag{36}$$

$$\geq \mathbb{E}[(\tau(\epsilon) \wedge n) - 1] \frac{\epsilon}{m}$$

Here we used that

$$\sum_{t=1}^{n-1} 1_{t \equiv 1 \ (\mathrm{mod}\ m)} = \left\lceil \frac{n-1}{m} \right\rceil \geq \frac{n-1}{m}$$

Combining equation 27, equation 30, equation 31, equation 32, equation 35 and equation 36 results in

$$\frac{\eta\epsilon}{2m}\left(\mathbb{E}[\tau(\epsilon)\wedge n]-1\right)\leq\frac{\eta}{2}G^2d_1(\mu_V,\mu_T)^2\left(\frac{\mathbb{E}[\tau(\epsilon)\wedge n]}{m}+1\right)+f_T(x_1)-f^*$$
$$+L\eta^2\sigma_v^2\mathbb{E}\left[\tau\left(\epsilon\right)\wedge n\right]+\eta\frac{\beta}{1-\alpha}\left(\mathbb{E}[\tau(\epsilon)\wedge n]+1\right).$$

This can be rearranged into

$$\left(\frac{\eta\epsilon}{m}-2L\eta^2\sigma_v^2-\frac{2\eta\beta}{1-\alpha}-\frac{\eta}{m}G^2d_1(\mu_V,\mu_T)^2\right)\mathbb{E}[\tau(\epsilon)\wedge n]\leq\eta G^2d_1(\mu_V,\mu_T)^2$$
$$+2(f_T(x_1)-f^*)+2\eta\frac{\beta}{1-\alpha}+\frac{\eta\epsilon}{m}.$$

which in turn is equivalent to

$$\mathbb{E}[\tau(\epsilon)\wedge n]\leq\frac{\eta G^2d_1(\mu_V,\mu_T)^2+2(f_T(x_1)-f^*)+\eta\epsilon/m+2\eta\beta/(1-\alpha)}{\eta\epsilon/m-2L\eta^2\sigma_v^2-2\eta\beta/(1-\alpha)-\eta G^2d_1(\mu_V,\mu_T)^2/m}. \tag{37}$$

Note that the sequence of random variables $\{(\tau(\epsilon)\wedge n)\}_{n=1,2,\dots}$ is monotone increasing, and converges pointwise to $\tau(\epsilon)$. Then the claimed relation equation 9 follows from equation 37 by the monotone convergence theorem.

Finally, by applying equation 28 with the roles of $f_V$ and $f_T$ reversed, and using the definition of $\tau(\epsilon)$, it follows that

$$\|\nabla f_T(x_{\tau(\epsilon)})\|\leq 2\|\nabla f_V(x_{\tau(\epsilon)})\|^2+2G^2d_1(\mu_V,\mu_T)^2$$
$$\leq 2\epsilon+2G^2d_1(\mu_V,\mu_T)^2.$$

$\square$

## PROOF OF COROLLARY 3.4

*Proof.* According to the assumption on the step-size $\eta$ (Equation equation 20),

$$\eta\left[(\epsilon-G^2d_1(\mu_V,\mu_T)^2)/m-\eta(2L\sigma_v^2+2R/(1-\alpha))\right]\geq\eta(1-c)(\epsilon-G^2d_1(\mu_V,\mu_T)^2)/m \tag{38}$$

and

$$\frac{1}{\eta}\leq\frac{L}{c}+\frac{m(2L\sigma_v^2+2R/(1-\alpha))}{c\left(\epsilon-G^2d_1(\mu_V,\mu_T)^2\right)}. \tag{39}$$

Combining these inequalities with the conclusion of Proposition 3.3 (relation equation 9) yields

$$\mathbb{E}[\tau(\epsilon)]\overset{\mathbf{A}}{\leq}\frac{\eta G^2d_1(\mu_V,\mu_T)^2+2(f_T(x_1)-f^*)+\eta\epsilon/m+2\eta\beta/(1-\alpha)}{\eta(\epsilon-G^2d_1(\mu_V,\mu_T)^2)/m-2L\eta^2\sigma_v^2-2\eta\beta/(1-\alpha)}.$$
$$\overset{\mathbf{B}}{=}\frac{\eta G^2d_1(\mu_V,\mu_T)^2+2(f_T(x_1)-f^*)+\eta\epsilon/m+2\eta^2R/(1-\alpha)}{\eta(\epsilon-G^2d_1(\mu_V,\mu_T)^2)/m-2L\eta^2\sigma_v^2-2\eta^2R/(1-\alpha)}. \tag{40}$$
$$\overset{\mathbf{C}}{\leq}\frac{\eta G^2d_1(\mu_V,\mu_T)^2+2(f_T(x_1)-f^*)+\eta\epsilon/m+2\eta^2R/(1-\alpha)}{\eta(1-c)\left(\epsilon-G^2d_1(\mu_V,\mu_T)^2\right)/m}.$$

Step **A** was established by Proposition 3.3. Step **B** uses the assumption that $\beta=\eta R$. Step **C** is an application of equation 38. Next, we will upper-bound the final inequality in three steps. First, using Inequality equation 39, we see that

$$\frac{2(f_T(x_1)-f^*)}{\eta(1-c)\left(\epsilon-G^2d_1(\mu_V,\mu_T)^2\right)/m}\leq\frac{2(f_T(x_1)-f^*)}{(1-c)\left(\epsilon-G^2d_1(\mu_V,\mu_T)^2\right)/m)}$$
$$\times\left(\frac{L}{c}+\frac{m(2L\sigma_v^2+2R/(1-\alpha))}{c\left(\epsilon-G^2d_1(\mu_V,\mu_T)^2\right)}\right)$$
$$=\frac{2m(f_T(x_1)-f^*)}{(1-c)\,c\left(\epsilon-G^2d_1(\mu_V,\mu_T)^2\right)}L$$
$$+\frac{2m^2(f_T(x_1)-f^*)}{(1-c)\,c\left(\epsilon-G^2d_1(\mu_V,\mu_T)^2\right)^2}\left(2L\sigma_v^2+2R/(1-\alpha)\right). \tag{41}$$

Next,

$$
\begin{aligned}
\frac{2\eta^2 R/(1-\alpha)}{\eta(1-c)\left(\epsilon - G^2 d_1(\mu_V, \mu_T)^2\right)/m} &= \frac{2\eta R/(1-\alpha)}{(1-c)\left(\epsilon - G^2 d_1(\mu_V, \mu_T)^2\right)/m} \\
&\leq \frac{2cR/(1-\alpha)}{(1-c)\left(\epsilon - G^2 d_1(\mu_V, \mu_T)^2\right)/m} \times \frac{\epsilon - G^2 d_1(\mu_V, \mu_T)^2}{m(2L\sigma_v^2 + 2R/(1-\alpha))} \\
&= \frac{c}{(1-c)} \times \frac{2R/(1-\alpha)}{(2L\sigma_v^2 + 2R/(1-\alpha))} \\
&\leq \frac{c}{(1-c)}.
\end{aligned}
\tag{42}
$$

Finally,

$$
\begin{aligned}
\frac{\eta G^2 d_1(\mu_V, \mu_T)^2 + \eta\epsilon/m}{\eta(1-c)\left(\epsilon - G^2 d_1(\mu_V, \mu_T)^2\right)/m} &= \frac{G^2 d_1(\mu_V, \mu_T)^2 + \epsilon/m}{(1-c)\left(\epsilon - G^2 d_1(\mu_V, \mu_T)^2\right)/m} \\
&= \frac{mc G^2 d_1(\mu_V, \mu_T)^2 + c\epsilon}{(1-c)c(\epsilon - G^2 d_1(\mu_V, \mu_T)^2)}.
\end{aligned}
\tag{43}
$$

Combining equation 40 with equation 41, equation 42 and equation 43, we find that

$$
\begin{aligned}
\mathbb{E}[\tau(\epsilon)] &\leq \frac{4m^2(f_T(x_1) - f^*)\left(L\sigma_v^2 + R/(1-\alpha)\right)}{(1-c)\,c\,(\epsilon - G^2 d_1(\mu_V, \mu_T)^2)^2} \\
&+ \frac{2Lm(f_T(x_1) - f^*) + mc G^2 d_1(\mu_V, \mu_T)^2 + c\epsilon}{(1-c)\,c\,(\epsilon - G^2 d_1(\mu_V, \mu_T)^2)} + \frac{c}{1-c}.
\end{aligned}
\tag{44}
$$

$\square$

## PROOF OF COROLLARY 3.5

*Proof.* If the algorithm runs until iteration $\tau(\epsilon)$, then the number of times that the full gradient of $f_V$ is calculated is $\lceil \tau(\epsilon)/m \rceil \leq \tau(\epsilon)/m + 1$, and the number of IFO calls for the training function is $\tau(\epsilon) - 1$. Therefore

$$
\mathrm{IFO}(\epsilon) \leq \left(\frac{\tau(\epsilon)}{m} + 1\right) n_V + (\tau(\epsilon) - 1) \leq \tau(\epsilon)\left(\frac{n_V}{m} + 1\right) + n_V.
\tag{45}
$$

Note that under our assumption on the gradient estimates $v_t$, we are in the unbiased setting where $R = 0$. Combining equation 44 with $R = 0$ and equation 45, we obtain

$$
\begin{aligned}
\mathbb{E}[\mathrm{IFO}(\epsilon)] &\leq \left(\frac{4m^2(f_T(x_1) - f^*)L\sigma_v^2}{(1-c)\,c\,(\epsilon - G^2 d_1(\mu_V, \mu_T)^2)^2}\right. \\
&+ \left.\frac{2Lm(f(x_1) - f^*) + mc G^2 d_1(\mu_V, \mu_T)^2 + c\epsilon}{(1-c)\,c\,(\epsilon - G^2 d_1(\mu_V, \mu_T)^2)} + \frac{c}{1-c}\right) \times \left(\frac{n_V}{m} + 1\right) + n_V.
\end{aligned}
\tag{46}
$$

Using $c = 1/2$ and neglecting lower-order terms, then,

$$
\mathbb{E}[\mathrm{IFO}(\epsilon)] = \mathcal{O}\left(\frac{mn_V + m^2}{(\epsilon - G^2 d_1(\mu_V, \mu_T)^2)^2} + n_V\right).
$$

$\square$

## C  ANALYSIS OF STACKED SGD

## PROOF OF PROPOSITION 4.2

*Proof.* Note that the variables inside the stacked algorithm satisfy the following identities: For all $1 \leq i \leq MK$,

$$
\widehat{x}_t^{c(i)} = x_t^i,
\tag{47}
$$

and for all pairs communication nodes nodes $1 \le j \le M$, and $1 \le k \le M$, and $t \ge 1$,

$$\widehat{x}_{t+\frac{1}{2}}^j = \widehat{x}_{t+\frac{1}{2}}^k. \tag{48}$$

Let $1 \le i \le M$ and $1 \le j \le M$ be arbitrary indices of communication nodes. By the definitions in Algorithm 2, we can see that for any $t \ge 1$,

$$
\begin{aligned}
&\|\widehat{x}_{t+1}^i - \widehat{x}_{t+1}^j\| \\
&\overset{\mathbf{A}}{=} \left\| \frac{1}{K+1} \left( \sum_{k \in c^{-1}(i)} x_{t+\frac{1}{2}}^k \right) - \frac{1}{K+1} \left( \sum_{k \in c^{-1}(j)} x_{t+\frac{1}{2}}^k \right) \right\| \\
&\overset{\mathbf{B}}{=} \left\| \frac{1}{K+1} \left( \sum_{k \in c^{-1}(i)} (x_t^k - \eta v_t^k) \right) - \frac{1}{K+1} \left( \sum_{k \in c^{-1}(j)} (x_t^k - \eta v_t^k) \right) \right\| \\
&\overset{\mathbf{C}}{=} \left\| \frac{1}{K+1} \left( \sum_{k \in c^{-1}(i)} (\widehat{x}_t^i - \eta v_t^k) \right) - \frac{1}{K+1} \left( \sum_{k \in c^{-1}(j)} (\widehat{x}_t^j - \eta v_t^k) \right) \right\| \\
&\overset{\mathbf{D}}{=} \left\| \frac{K}{K+1} \left( \widehat{x}_t^i - \widehat{x}_t^j \right) - \frac{1}{K+1} \sum_{k \in c^{-1}(i)} \eta v_t^k + \frac{1}{K+1} \sum_{k \in c^{-1}(j)} \eta v_t^k \right\| \\
&\overset{\mathbf{E}}{\le} \frac{K}{K+1} \left\| \widehat{x}_t^i - \widehat{x}_t^j \right\| + \eta \left\| \frac{1}{K+1} \sum_{k \in c^{-1}(i)} v_t^k - \frac{1}{K+1} \sum_{k \in c^{-1}(j)} v_t^k \right\|.
\end{aligned}
\tag{49}
$$

Step $\mathbf{A}$ follows from the definition of $\widehat{x}_{t+1}^i, \widehat{x}_{t+1}^j$ in Algorithm 2 and Equation equation 48. Step $\mathbf{B}$ follows from the definition of $x_{t+\frac{1}{2}}^k$ in Algo. 2. Step $\mathbf{C}$ follows from Equation equation 47. Step $\mathbf{D}$ follows from rearranging terms in the previous step, and noting that there are $K$ workers in a cluster. Finally, Step $\mathbf{E}$ is simply the triangle inequality.

For the second term on the right of the final inequality above,

$$
\begin{aligned}
&\left\| \frac{1}{K+1} \sum_{k \in c^{-1}(i)} v_t^k - \frac{1}{K+1} \sum_{k \in c^{-1}(j)} v_t^k \right\| \\
&= \frac{K}{K+1} \left\| \nabla f(\widehat{x}_t^i) - \nabla f(\widehat{x}_t^j) \right. \\
&\quad + \left( \frac{1}{K} \sum_{k \in c^{-1}(i)} v_t^k - \nabla f_T(\widehat{x}_t^i) \right) - \left( \frac{1}{K} \sum_{k \in c^{-1}(j)} v_t^k - \nabla f_T(\widehat{x}_t^j) \right) \right\| \\
&\le \frac{KL}{K+1} \|\widehat{x}_t^i - \widehat{x}_t^j\| \\
&\quad + \frac{K}{K+1} \left\| \left( \frac{1}{K} \sum_{k \in c^{-1}(i)} v_t^k - \nabla f_T(\widehat{x}_t^i) \right) - \left( \frac{1}{K} \sum_{k \in c^{-1}(j)} v_t^k - \nabla f_T(\widehat{x}_t^j) \right) \right\|.
\end{aligned}
\tag{50}
$$

where the last inequality uses the Lipschitz gradient property (Assumption 2.1) and the triangle inequality. Combining equation 49 and equation 50 yields

$$
\begin{aligned}
\left\| \widehat{x}_{t+1}^i - \widehat{x}_{t+1}^j \right\| &\le \frac{K}{K+1}(1+L\eta) \left\| \widehat{x}_t^i - \widehat{x}_t^j \right\| \\
&\quad + \frac{\eta K}{K+1} \left\| \left( \frac{1}{K} \sum_{k \in c^{-1}(i)} v_t^k - \nabla f_T(\widehat{x}_t^i) \right) - \left( \frac{1}{K} \sum_{k \in c^{-1}(j)} v_t^k - \nabla f_T(\widehat{x}_t^j) \right) \right\|.
\end{aligned}
\tag{51}
$$

Note that for any $k_1 > 0$ and all $a, b$ we have

$$|a + b|^2 \le (1 + k_1)a^2 + \left(1 + \frac{1}{k_1}\right) b^2. \tag{52}$$

Combining equation 51 and equation 52 while using the assumption that $\eta \leq 1/(2LK)$ results in

$$
\begin{aligned}
\|\widehat{x}_{t+1}^i - \widehat{x}_{t+1}^j\|^2 &\leq \frac{(1+k_1)K^2}{(K+1)^2}\left(1 + \frac{1}{2K}\right)^2 \left\|\widehat{x}_t^i - \widehat{x}_t^j\right\|^2 \\
&+ \eta\frac{(1+k_1)K}{k_1(K+1)^2}\frac{1}{2L}\left\|\left(\frac{1}{K}\sum_{k\in c^{-1}(i)} v_t^k - \nabla f_T(\widehat{x}_t^i)\right) - \left(\frac{1}{K}\sum_{k\in c^{-1}(j)} v_t^k - \nabla f_T(\widehat{x}_t^j)\right)\right\|^2.
\end{aligned}
\tag{53}
$$

Let $k_1 = \frac{(K+3/4)^2}{(K+1/2)^2} - 1$. Then by the definition of $\alpha$,

$$
\frac{(1+k_1)K^2}{(K+1)^2}\left(1 + \frac{1}{2K}\right)^2 = \frac{(1+k_1)(K+1/2)^2}{(K+1)^2} = \frac{(K+3/4)^2}{(K+1/2)^2} = \alpha.
\tag{54}
$$

Furthermore,

$$
\frac{1+k_1}{k_1} \leq \frac{(K+1)^2}{K/2 + 5/16}
$$

so

$$
\begin{aligned}
\eta\frac{(1+k_1)K}{k_1(K+1)^2}\frac{1}{2L} &\leq \eta\frac{(K+1)^2}{(K/2+5/16)}\frac{K}{(K+1)^2}\frac{1}{2L} \\
&= \eta\frac{K}{(K/2+5/16)}\frac{1}{2L} \\
&= \eta\frac{K}{L(K+5/8)}.
\end{aligned}
\tag{55}
$$

Multiplying each side of equation 53 by $L^2/M^2$, summing the resulting inequality over $i = 1, \ldots, M$ and $j = 1, \ldots, M$, and using the relations equation 54, equation 55 we see that for all $t \geq 1$,

$$
V_{t+1} \leq \alpha V_t + U_t.
$$

It remains to confirm that $\mathbb{E}[U_t \mid \mathcal{F}_{t-1}]$ for all $t \geq 1$.

Taking expectations in equation 14b, while noting that $\widehat{x}_t^{c(k)} = x_t^k$ and applying the variance bound equation 13 along with the inequality $\|a+b\|^2 \leq 2\left(\|a\|^2 + \|b\|^2\right)$ it holds for all $t \geq 1$ that

$$
\begin{aligned}
&\mathbb{E}\left[U_t \mid \mathcal{F}_{t-1}\right] \\
&\leq \eta \cdot \frac{KL}{(K+5/8)} \\
&\quad \times \frac{1}{M^2}\sum_{i=1}^M\sum_{j=1}^M \mathbb{E}\left[\left\|\left(\frac{1}{K}\sum_{k\in c^{-1}(i)} v_t^k - \nabla f_T(\widehat{x}_t^i)\right) - \left(\frac{1}{K}\sum_{k\in c^{-1}(j)} v_t^k - \nabla f_T(\widehat{x}_t^j)\right)\right\|^2\,\middle|\,\mathcal{F}_{t-1}\right] \\
&= \eta\frac{KL}{(K+5/8)}\frac{1}{M^2}\sum_{i=1}^M\sum_{j=1}^M 2\left(\frac{\sigma_v^2}{K} + \frac{\sigma_v^2}{K}\right) \\
&= \beta.
\end{aligned}
$$

$\square$

## PROOF OF PROPOSITION 4.3

*Proof.* Note that according to Line 9 of SSGD,

$$
\widehat{x}_t = \widehat{x}_{t+\frac{1}{2}}^1 = \ldots = \widehat{x}_{t+\frac{1}{2}}^M
\tag{56}
$$

and Lines 7 and 10 mean that for all $j$ with $c(j) = i$,

$$
\widehat{x}_t^i = x_t^j
\tag{57}
$$

Using these equations, together with the definitions in the SSGD algorithm, we obtain that

$$
\begin{aligned}
\widehat{x}_{t+1} &\stackrel{\mathbf{A}}{=} \frac{1}{M} \sum_{i=1}^{M} \frac{1}{K+1} \left( \widehat{x}_{t+\frac{1}{2}}^{i} + \sum_{j \in c^{-1}(i)} x_{t+\frac{1}{2}}^{j} \right) \\
&\stackrel{\mathbf{B}}{=} \left( \frac{1}{K+1} \widehat{x}_t + \frac{1}{M} \sum_{i=1}^{M} \frac{1}{K+1} \sum_{j \in c^{-1}(i)} \left( x_t^j - \eta v_t^j \right) \right) \\
&\stackrel{\mathbf{C}}{=} \left( \frac{1}{K+1} \widehat{x}_t + \frac{1}{M} \sum_{i=1}^{M} \frac{1}{K+1} \sum_{j \in c^{-1}(i)} \left( \widehat{x}_t^i - \eta v_t^j \right) \right) \\
&\stackrel{\mathbf{D}}{=} \left( \frac{1}{K+1} \widehat{x}_t + \frac{K}{K+1} \widehat{x}_t - \frac{1}{M} \sum_{i=1}^{M} \frac{1}{K+1} \sum_{j \in c^{-1}(i)} \eta v_t^j \right) \\
&\stackrel{\mathbf{E}}{=} \widehat{x}_t - \eta \left( \frac{1}{M} \sum_{i=1}^{M} \frac{1}{K+1} \sum_{j \in c^{-1}(i)} v_t^j \right).
\end{aligned}
$$

For Step **A**, note that $\widehat{x}_t$ is the average of the $\widehat{x}_t^i$, and then use definition of the $\widehat{x}_t^i$ from Line 9 of the SSGD algorithm. Step **B** follows from equation 56 and the definition of the $x_{t+\frac{1}{2}}^i$ from Line 5 of SSGD. Step **C** follows from equation 57. Step **D** follows by rearranging terms in the previous step, and again noting the definition of $\widehat{x}_t$ is the average of the $\widehat{x}_t^i$. Step **E** follows by grouping terms.

Continuing, we can express this recursion as an approximate form of gradient descent:

$$
\begin{aligned}
\widehat{x}_{t+1} &= \widehat{x}_t - \eta \left( \frac{1}{M} \sum_{i=1}^{M} \frac{1}{K+1} \sum_{j \in c^{-1}(i)} v_t^j \right) \\
&= \widehat{x}_t - \eta \frac{K}{K+1} \left( \frac{1}{M} \sum_{i=1}^{M} \frac{1}{K} \sum_{j \in c^{-1}(i)} v_t^j \right) \\
&= \widehat{x}_t - \frac{\eta K}{K+1} \left( v_t + \Delta_t \right),
\end{aligned}
$$

where $v_t$ and $\Delta_t$ are

$$
v_t = \nabla f_T(\widehat{x}_t) + \frac{1}{M} \sum_{i=1}^{M} \left( \frac{1}{K} \left( \sum_{j \in c^{-1}(i)} v_t^j \right) - \nabla f_T(\widehat{x}_t^i) \right) \tag{58}
$$

and

$$
\Delta_t = \frac{1}{M} \left( \sum_{i=1}^{M} \nabla f_T(\widehat{x}_t^i) \right) - \nabla f_T(\widehat{x}_t). \tag{59}
$$

Based on the definition of $v_t$ in Equation equation 58 and on Assumption 4.1, for all $t \geq 1$ it holds that

$$
\mathbb{E}\left[ v_t - \nabla f_T(\widehat{x}_t) \mid \mathcal{F}_{t-1} \right] = 0,
$$

$$
\mathbb{E}\left[ \| v_t - \nabla f_T(\widehat{x}_t) \|^2 \mid \mathcal{F}_{t-1} \right] \leq \frac{\sigma_v^2}{K}.
$$

Thus Assumption 3.1 is confirmed. Next we consider the Lyapunov condition of Assumption 3.2. Let the variables $U_t, V_t$ for $t \geq 1$ and the constants $\alpha, \beta$ be defined as in equation 14a-equation 14d.

Then by Assumption 2.1,

$$\|\Delta_t\|^2 = \left\| \frac{1}{M} \sum_{i=1}^{M} \nabla f_T(\widehat{x}_t^i) - \nabla f(\widehat{x}_t) \right\|^2$$

$$\leq L^2 \left\| \frac{1}{M} \sum_{i=1}^{M} (\widehat{x}_t^i - \widehat{x}_t) \right\|^2$$

$$= L^2 \left\| \frac{1}{M} \sum_{i=1}^{M} \left( \widehat{x}_t^i - \frac{1}{M} \sum_{j=1}^{M} \widehat{x}_t^j \right) \right\|^2$$

$$= L^2 \left\| \frac{1}{M^2} \sum_{i=1}^{M} \sum_{j=1}^{M} \left( \widehat{x}_t^i - \widehat{x}_t^j \right) \right\|^2$$

$$\leq L^2 \frac{1}{M^2} \sum_{i=1}^{M} \sum_{j=1}^{M} \left\| \widehat{x}_t^i - \widehat{x}_t^j \right\|^2$$

$$= V_t.$$

The second line uses the Lipschitz gradient property, and the second to last line follows from Jensen's inequality (Section 6.6, Williams (1991)).

By our assumption that each node has the same initial state, then $V_1 = 0$, hence Inequality equation 6 holds. The Inequalities equation 7 and equation 8 are established by Prop 4.1. According to the definition of $\beta$ (Equation equation 14d, we may write $\beta = \eta R$ where $R = 4L\sigma_v^2/(K + 5/8)$.

Note that $1/(1 - \alpha) = 2(K + 1)^2/(K + 5/8)$. Therefore,

$$\frac{R}{1 - \alpha} = 8L\sigma_v^2(K + 1)^2/(K + 5/8)^2 \leq 32L\sigma_v^2 \tag{60}$$

According to Corollary 3.4, then, a step-size of

$$\eta = c \cdot \min \left\{ \frac{1}{L}, \frac{\epsilon - G^2 d_1(\mu_V, \mu_T)^2}{m(2L\sigma_v^2/K + 16L\sigma_v^2(K + 1)^2/(K + 5/8)^2)} \right\}$$

leads to

$$\mathbb{E}[\tau(\epsilon)] \leq \frac{4m^2(f_T(x_1) - f^*)\left(L\sigma_v^2/K + 32L\sigma_v^2\right)}{(1 - c)\,c\,(\epsilon - G^2 d_1(\mu_V, \mu_T)^2)^2}$$
$$+ \frac{2Lm(f_T(x_1) - f^*) + mcG^2 d_1(\mu_V, \mu_T)^2 + c\epsilon}{(1 - c)\,c\,(\epsilon - G^2 d_1(\mu_V, \mu_T)^2)} + \frac{c}{1 - c}. \tag{61}$$

Dropping the lower order terms, we see that

$$\mathbb{E}[\tau(\epsilon)] = \mathcal{O}\left( \frac{m^2}{(1 - c)c(\epsilon - G^2 d_1(\mu_V, \mu_T)^2)^2} \right). \tag{62}$$

$\square$

## PROOF OF COROLLARY 4.4

*Proof.* If SSGD runs until iteration $\tau(\epsilon)$, then number of times that the full gradient of $f_V$ is calculated is $\lceil \tau(\epsilon)/m \rceil \leq \tau(\epsilon)/m + 1$, and the number of IFO calls for the training function is $(\tau(\epsilon) - 1)K$. Therefore

$$\text{IFO}(\epsilon) \leq \left( \frac{\tau(\epsilon)}{m} + 1 \right) n_V + (\tau(\epsilon) - 1)K \leq \tau(\epsilon)\left( \frac{n_V}{m} + K \right) + n_V. \tag{63}$$

Next, note that $(1 - c)c = (1 - \frac{1}{4K})\frac{1}{4K} = \frac{4K-1}{(4K)^2}$, which implies

$$\frac{1}{(1 - c)c} = \frac{16K^2}{4K - 1} \leq \frac{16K}{3}. \tag{64}$$

Combining equation 62, equation 63, and equation 64 we see that

$$\mathbb{E}\left[\text{IFO}(\epsilon)\right] = \mathcal{O}\left(\frac{mK(n_V + mK)}{(\epsilon - G^2 d_1(\mu_V, \mu_T)^2)^2} + n_V\right).$$

□

## D  ANALYSIS OF DECENTRALIZED SGD

The following result is a restatement of Lemma 5 of Lian et al. (2017).

**Lemma D.1.** *Under Assumption 5.1, the matrix* $\lim_{k\to\infty} a^k$ *is well-defined and has entries* $a_{i,j}^\infty = \frac{1}{M}$ *for* $1 \le i, j \le M$. *Furthermore, for all* $k \ge 1$, *bound on the spectral gap implies* $\|a^\infty - a^k\|^2 \le \rho^k$.

PROOF OF PROPOSITION 5.3

*Proof.* For ease of notation, for each $t \ge 1$ let $y_t$ and $z_t$ be the $M$-dimensional vectors with components defined as

$$y_t^i = x_t^i - \overline{x}_t, \tag{65}$$

$$z_t^i = v_t^i - \frac{1}{M}\sum_{j=1}^M v_t^j. \tag{66}$$

Let $a^\infty$ be the $M \times M$ matrix with entries $a_{i,j}^\infty = \frac{1}{M}$ (see Lemma D.1). Then according to Line 5 of Algorithm 3, the $y_t$ satisfy the recursion

$$y_{t+1} = (a - a^\infty)y_t + \eta z_t. \tag{67}$$

Note that $z_t$ can be expressed as

$$
\begin{aligned}
z_t^i = {}& \nabla f(x_t^i) - \nabla f(\overline{x}_t) \\
& + v_t^i - \nabla f(x_t^i) - \frac{1}{M}\sum_{j=1}^M (v_t^j - \nabla f(x_t^j)) + \frac{1}{M}\sum_{j=1}^M (\nabla f(x_t^j) - \nabla f(\overline{x}_t))
\end{aligned} \tag{68}
$$

Using the Lipschitz property (Assumption 2.1 ) then,

$$|z_n^i| \le L|x_n^i - \overline{x}_n| + |v_n^i - \nabla f(x_n^i)| + \frac{1}{M}\sum_{j=1}^M |v_n^j - \nabla f(x_n^j)| + \frac{L}{M}\sum_{j=1}^M |x_n^j - \overline{x}_n| \tag{69}$$

Squaring and summing Equation equation 69,

$$
\begin{aligned}
\sum_{i=1}^M |z_n^i|^2 ={}& \sum_{i=1}^M \left(L|x_n^i - \overline{x}_n| + |v_n^i - \nabla f(x_n^i)| + \frac{1}{M}\sum_{j=1}^M |v_n^j - \nabla f(x_n^j)| + \frac{L}{M}\sum_{j=1}^M |x_n^j - \overline{x}_n|\right)^2 \\
={}& L^2 4\sum_{i=1}^M |x_n^i - \overline{x}_n|^2 + 4\sum_{i=1}^M |v_n^i - \nabla f(x_n^i)|^2 \\
& + \frac{4}{M}\sum_{i=1}^M\sum_{j=1}^M |v_n^j - \nabla f(x_n^j)|^2 + \frac{4L^2}{M}\sum_{i=1}^M\sum_{j=1}^M |x_n^j - \overline{x}_n|^2 \\
={}& L^2 8\|y_n\|^2 + 8\sum_{i=1}^M |v_n^i - \nabla f(x_n^i)|^2
\end{aligned}
$$

Taking square roots on each sides of this equation yields

$$
\begin{aligned}
\|z_n\| &\le \sqrt{L^2 8\|y_n\|^2 + 8\sum_{i=1}^M |v_n^i - \nabla f(x_n^i)|^2} \\
&\le \sqrt{L^2 8}\|y_n\| + \sqrt{8\sum_{i=1}^M |v_n^i - \nabla f(x_n^i)|^2}
\end{aligned} \tag{70}
$$

Combining equation 67 and equation 70, then,

$$\|y_{n+1}\| \le \left( \|a - a^\infty\| + \eta L \sqrt{8} \right) \|y_n\| + \eta \sqrt{8 \sum_{i=1}^{M} |v_n^i - \nabla f(x_n^i)|^2}$$

$$\le \left( \sqrt{\rho} + \eta L \sqrt{8} \right) \|y_n\| + \eta \sqrt{8 \sum_{i=1}^{M} |v_n^i - \nabla f(x_n^i)|^2}$$

In the second step we have applied Assumption equation 5.1 and Lemma D.1. Squaring this equation, for any $k_1 > 0$ it holds that

$$\|y_{n+1}\|^2 \le (1 + k_1) \left( \sqrt{\rho} + \eta L \sqrt{8} \right)^2 \|y_n\|^2 + \eta^2 \left( 1 + \frac{1}{k_1} \right) 8 \sum_{i=1}^{M} |v_n^i - \nabla f(x_n^i)|^2. \qquad (71)$$

Let $k_1 = \frac{1-\rho}{2\rho}$ (in which case $1 + \frac{1}{k_1} = \frac{1+\rho}{1-\rho}$). Multiplying each side of 71 by $L^2/M$ and noting that $V_t = \frac{L^2}{M} \|y_t\|^2$, it follows that

$$V_{t+1} \le \frac{(1+\rho)}{2\rho} \left( \sqrt{\rho} + \eta L \sqrt{8} \right)^2 V_t + U_t \qquad (72)$$

It follows from the variance bound in Assumption 5.2 that

$$\mathbb{E}\left[ U_t \mid \mathcal{F}_{t-1} \right] \le 8\,\eta^2\, L^2 \frac{1+\rho}{1-\rho} \sigma_v^2 \qquad (73)$$

Using the assumption that $\eta \le \frac{1-\sqrt{\rho}}{4L\sqrt{2}}$, then equation 72 and equation 73 become, respectively,

$$V_{t+1} \le \frac{(1+\rho)}{2\rho} \left( \frac{\sqrt{\rho}+1}{2} \right)^2 V_t + U_t$$

and

$$\mathbb{E}\left[ U_t \mid \mathcal{F}_{t-1} \right] \le \sqrt{2}\,\eta(1 - \sqrt{\rho})\, L \frac{1+\rho}{1-\rho} \sigma_v^2$$

$$\le \eta \frac{L\sqrt{2}}{1-\rho} \sigma_v^2$$

$\square$

PROOF OF PROPOSITION 5.4

*Proof.* To begin, note that the system average $\overline{x}_t$ satisfies the recursion

$$\overline{x}_{t+1} = \overline{x}_t + \frac{\eta}{M} \sum_{i=1}^{M} v_t^i \qquad (74)$$

Define the variables $v_t$ and $\Delta_t$, for $t \ge 1$, as

$$v_t = \nabla f(\overline{x}_t) + \frac{1}{M} \sum_{i=1}^{M} \left( v_t^i - \nabla f(x_t^i) \right)$$

$$\Delta_n = \frac{1}{M} \sum_{i=1}^{M} \left( \nabla f(x_t^i) - \nabla f(\overline{x}_t) \right)$$

Then we can express the recursion equation 74 as

$$\overline{x}_{t+1} = \eta \left( v_t + \Delta_t \right)$$

We will show that this can be interpreted as a form of biased SGD and therefore we may apply Corollary 3.4. For the unbiased component $v_t$, observe that

$$\mathbb{E}\left[v_t - \nabla f_T(x_t) \mid \mathcal{F}_{t-1}\right] = \mathbb{E}\left[\frac{1}{M}\sum_{i=1}^{M}(v_t^i - \nabla f(x_t^i)) \mid \mathcal{F}_{t-1}\right] = 0 \tag{75}$$

and

$$\mathbb{E}\left[\|v_t - \nabla f_T(x_t)\|^2 \mid \mathcal{F}_{t-1}\right] \le \mathbb{E}\left[\frac{1}{M}\sum_{i=1}^{M}|v_t^i - \nabla f(x_t^i)|^2\right] = \sigma_v^2 \tag{76}$$

For the bias term, note that

$$\|\Delta_t\|^2 \le \frac{L^2}{M}\sum_{i=1}^{M}|x_t^i - \overline{x}_t|^2 = V_t$$

Assumption equation 3.1 follows from equation 75 and equation 76, while Assumption equation 3.2 follows from Proposition 5.3. The result then follows from Corollary 3.4. □

# E  ANALYSIS OF SVRG

For the analysis of SVRG, define the filtration $\{\mathcal{F}_t\}_{t=0,1,\dots}$ as follows. $\mathcal{F}_0 = \sigma(x_m^1)$ and for all $s \ge 1$,

$$\mathcal{F}_s = \sigma\left(\{x_m^1\} \cup \left\{i_t^j \,\middle|\, 0 \le t \le m-1, 1 \le j \le s\right\}\right).$$

We will leverage some prior results concerning the behavior of SVRG. The following is adapted from Reddi et al. (2016a).

**Proposition E.1.** *Let Assumptions 2.1 and 2.2 hold. Let $\beta > 0$ and define the constants $c_m, c_{m-1}, \dots, c_0$ as follows: $c_m = 0$, and for $0 \le t \le m-1$, let $c_t = c_{t+1}(1+\eta\beta+2\eta^2 L^2)+\eta^2 L^3$. Define $\Gamma_t$ for $0 \le t \le m-1$ as $\Gamma_t = \eta - \frac{c_{t+1}\eta}{\beta} - \eta^2 L - 2c_{t+1}\eta^2$. Suppose that the step-size $\eta$ and the analysis constant $\beta$ are chosen so that $\Gamma_t > 0$ for $0 \le t \le m-1$, and set $\gamma = \inf_{0 \le t < m} \Gamma_t$. Then for all $s \ge 1$,*

$$\sum_{t=0}^{m-1} \mathbb{E}[\|\nabla f_T(x_t^{s+1})\|^2 \mid \mathcal{F}_{s-1}] \le \frac{f_T(x_m^s) - \mathbb{E}[f_T(x_m^{s+1}) \mid \mathcal{F}_{s-1}]}{\gamma}. \tag{77}$$

*Furthermore, if $\eta$ is of the form $\eta = \xi/(Ln^{2/3})$ for some $\xi \in (0,1)$ and if the epoch length is set to $m = \lfloor n/(3\xi) \rfloor$, then there is a value for $\beta$ such that $\gamma \ge \frac{\nu(\xi)}{Ln^{2/3}}$ where $\nu(\xi)$ is a constant dependent only on $\xi$. In particular, if $\xi = 1/4$ then*

$$\gamma \ge \frac{1}{40Ln^{2/3}}. \tag{78}$$

*Proof.* The proof of equation 77 follows from nearly the same reasoning used to establish Equation 10 in (Section B, Reddi et al. (2016a)), the only difference being that conditional expectations replace expectations in all of the relevant formulas. The details are left to the reader.

Formula equation 78 is a numerical inequality whose proof can be derived from the proof of Theorem 3 given in in (Appendix B, Reddi et al. (2016a)). □

PROOF OF PROPOSITION 6.1

*Proof.* First, note that $\tau(\epsilon)$ is a well-defined stopping time with respect to the filtration $\{\mathcal{F}_s\}_{s=0,1,\dots}$. For $s = 1, 2, \dots$ define the random variables $\delta_s$ as

$$\delta_s = \sum_{t=0}^{m-1} \|\nabla f_T(x_t^{s+1})\|^2 - \frac{f_T(x_m^s) - f_T(x_m^{s+1})}{\gamma}$$

It holds trivially that for all $s \geq 1$,

$$\sum_{t=0}^{m-1} \|\nabla f_T(x_t^{s+1})\|^2 = \frac{f_T(x_m^s) - f_T(x_m^{s+1})}{\gamma} + \delta_s \tag{79}$$

and by Proposition E.1, for all $s \geq 1$,

$$\mathbb{E}[\delta_s \mid \mathcal{F}_{s-1}] = \sum_{t=0}^{m-1} \mathbb{E}\left[\|\nabla f_T(x_t^{s+1})\|^2 \mid \mathcal{F}_{s-1}\right] - \frac{f_T(x_m^s) - \mathbb{E}[f_T(x_m^{s+1}) \mid \mathcal{F}_{s-1}]}{\gamma} \tag{80}$$
$$\leq 0.$$

Summing Equation equation 79 over $s = 1, \ldots, q$ yields

$$\sum_{s=1}^{q} \sum_{i=0}^{m-1} \|\nabla f_T(x_i^{s+1})\|^2 = \frac{f_T(x_m^1) - f_T(x_m^{q+1})}{\gamma} + \sum_{s=1}^{q} \delta_s, \tag{81}$$

Rearranging terms and noting that $f_T(x_m^{q+1}) \geq f^*$ results in

$$\gamma \sum_{s=1}^{q} \sum_{i=0}^{m-1} \|\nabla f_T(x_i^{s+1})\|^2 \leq f_T(x_m^1) - f^* + \gamma \sum_{s=1}^{q} \delta_s. \tag{82}$$

It follows that

$$\gamma \sum_{s=1}^{q} \|\nabla f_T(x_0^{s+1})\|^2 \leq f_T(x_m^1) - f^* + \gamma \sum_{s=1}^{q} \delta_s. \tag{83}$$

For $r \geq 1$, let $\tau(\epsilon) \wedge r$ be the stopping time which is the minimum of $\tau(\epsilon)$ and the constant value $r$. Applying Proposition A.1 together with Equation 80, it holds that

$$\mathbb{E}\left[\sum_{s=1}^{\tau(\epsilon) \wedge r} \delta_s\right] \leq 0 \tag{84}$$

Furthermore, by definition of $\tau$,

$$\mathbb{E}\left[\sum_{s=1}^{\tau(\epsilon) \wedge r} \|\nabla f_T(x_0^{s+1})\|^2\right] \geq \mathbb{E}\left[\sum_{s=1}^{(\tau(\epsilon) \wedge r)-1} \|\nabla f_T(x_0^{s+1})\|^2\right] \geq \mathbb{E}\left[\sum_{s=1}^{(\tau(\epsilon) \wedge r)-1} \epsilon\right] \tag{85}$$
$$= \epsilon \mathbb{E}[(\tau(\epsilon) \wedge r) - 1].$$

Combining equation 83, equation 84, and equation 85 yields

$$\gamma \epsilon \mathbb{E}[(\tau(\epsilon) \wedge n) - 1] \leq f_T(x_m^1) - f^*$$

Rearranging terms in the above yields

$$\mathbb{E}[\tau(\epsilon) \wedge n] \leq \frac{f_T(x_m^1) - f^*}{\gamma \epsilon} + 1.$$

Applying the monotone convergence theorem, then,

$$\mathbb{E}[\tau(\epsilon)] \leq \frac{f_T(x_m^1) - f^*}{\gamma \epsilon} + 1.$$

Next, specialize $\eta$ and $m$ to $\eta = \xi/(Ln^{2/3})$ and $m = \lfloor n/(3\xi) \rfloor$ with $\xi = 1/4$. Then by Proposition E.1, $\gamma \geq 1/(40Ln^{2/3})$. Therefore,

$$\mathbb{E}[\tau(\epsilon)] \leq \frac{40Ln^{2/3}(f_T(x_m^1) - f^*)}{\epsilon} + 1.$$

$\square$

# F  GENERALIZATION ANALYSIS

## PROOF OF COROLLARY 7.2

*Proof.* Under our assumption on the Lipschitz property of $y \mapsto \nabla_x f(y, x)$, it holds that

$$\|\nabla f_\mu(x_{\tau(\epsilon)})\| \leq \|\nabla f_V(x_{\tau(\epsilon)})\| + G d_1(\mu, \mu_V).$$

Squaring and taking expectations, while noting that $d_1 \leq d_2$ (see Villani (2008), Remark 6.6),

$$\mathbb{E}[\|\nabla f_\mu(x_{\tau(\epsilon)})\|^2] \leq 2\mathbb{E}[\|\nabla f_V(x_{\tau(\epsilon)})\|^2] + 2G^2 \mathbb{E}[d_2(\mu, \mu_V)^2].$$

Using the Wasserstein concentration bound from Theorem 7.1 and the definition of $\tau(\epsilon)$, we obtain

$$\mathbb{E}[\|\nabla f_\mu(x_{\tau(\epsilon)})\|^2] \leq 2\epsilon + 2G^2 \kappa_d J n_V^{-3/d}.$$

$\square$

# G  EXPERIMENTAL METHODOLOGY

## G.1  SSGD EXPERIMENTS

The neural network model used for these experiments is LSTNet Lai et al. (2018)with CUDA-aware MPI and extended to use the Stacked SGD training method. The objective function is the squared error between the prediction and the true sensor measurement, averaged over the dataset of training instances.

Our experiments compared SSGD (Algorithm 2) and SGD (1). For SSGD, each cluster has 4 worker nodes and 1 communication node. Hence, $K = 4$ and $M$ is varied from 1 to 64. Since one physical node has four GPU devices, each physical node in the HPC environment is modeled a single local cluster in SSGD. The scalability of SSGD is compared with the basic parallel implementation of SGD, where all node synchronize with all-reduce collective communication call after each parameter update.

The experiment is conducted in a high performance computing environment that is equipped with 108 nodes. Each node has a dual Intel Xeon E5-2695v4 (Broadwell) CPU, four NVidia K40 GPUs, SAS-based local storage, and 256 GB of memory. The nodes are inter-connected with a non-blocking Infiniband EDR fabric.

## G.2  SVRG EXPERIMENTS

For the first experiment in this section, we trained a neural network with no hidden layer (i.e., a logistic classifier) for the MNIST classification task. For the second experiment, we trained a neural network with one hidden layer of $n = 100$ nodes on the CIFAR-10 classification task. The activation function in the hidden layer was the logistic function $\sigma(x) = (1 + e^{-x})^{-1}$. In both cases, the objective function is the average cross entropy loss.

Our experiments compared SVRG (Algorithm 4) and SGD (Algorithm 1). The only modification to the algorithms was that mini-batches were used to make the gradient estimates. For each algorithm we determined the values of the learning rate and mini-batch size using a validation set. For the learning rate, we searched over the values $\eta \in \{0.001, 0.01, 0.1, 1.0\}$. For the mini-batch size, we considered values in $\{32, 64, 128\}$. The values used for full training were determined by running the training procedure for several epochs and evaluating the resulting model on a held-out portion of the training dataset. The parameters that gave best performance on this held-out dataset were used for full training. Using the found settings, we ran five independent runs of training, and report the mean and confidence bands (representing one standard deviation) for the expected IFO calls.

