# OpenReview forum: "On the expected running time of nonconvex optimization with early stopping"
_ICLR.cc/2020/Conference — Reject_

### Official Review · AnonReviewer2 · 2019-10-21
**Official Blind Review #2**

**Rating:** 6

**Review:**

The paper studies the problem of the number of first-order-oracle calls for the SGD type of algorithms to find a stationary point of the objective function. The main results in the paper are built upon a new, general framework to analyze the SGD type of algorithms.


The main framework can be summarized as follows: At each iteration, the algorithm receives h_t, a (potentially biased) estimator of the gradient at a given point x_t, and performs a simple update x_{t + 1} = x_t - \eta * h_t. The framework says that as long as the norm (V_t) of \Delta_t = h_t - v_t (where v_t is an unbiased estimator of the true gradient with bounded variance) satisfies a particular Lyapunov-type inequality, then the algorithm can find an epsilon-stationary point as long as epsilon is not too small.


The analysis of the framework is quite standard, one only needs to write the decrement in function value at each iteration into the following three terms: the norm of the true gradient of the function, \delta_t: the difference between v_t and the true gradient (so E[\delta_t] = 0) and \Delta_t: the difference between the received gradient h_t and v_t.


The authors showed some application of this framework in Stacked SGD and decentralized SGD. The main intuitions of these applications are (1). \Delta_t comes from the synchronization difference of the nodes when computing the gradient. (2). The shrinking of V_t is due to the  (better) synchronization at each iteration. (3). The increment of V_t is due to the gradient update.

Overall, I find the general framework quite interesting and potentially useful for future research and could be used as a guide for choosing the proper algorithm in distributed computation.  The bounds in this paper are also in principle tight. The only question I have about this result is the dependency of m (the number of iterations between each evaluation of the gradient norm of the underlying function). (1). How can this (the evaluation of the gradient norm of the underlying function)) be done in a decentralized environment? What is the computation overhead?  (For example in DSGD, how can we compute \bar{x}_t?) (2). It seems that the computation cost (number of IFO) scales quadratically with respect to m. What is the intuition for this scaling? It appears to me that the scaling should be linear or better (the worst case is that within the "m" iterations, only one iteration has gradient >= epsilon). The authors should elaborate more on this point.




**Experience Assessment:**

I have published in this field for several years.

**Review Assessment: Checking Correctness Of Derivations And Theory:**

I assessed the sensibility of the derivations and theory.

**Review Assessment: Checking Correctness Of Experiments:**

I did not assess the experiments.

**Review Assessment: Thoroughness In Paper Reading:**

I read the paper thoroughly.

---

### Official Review · AnonReviewer1 · 2019-10-23
**Official Blind Review #1**

**Rating:** 3

**Review:**

In this paper, the authors consider stochastic optimization in the setting where a validation function is used to guide the termination of the algorithm. In more details, the algorithm terminates if the gradient of the validation function at an iterate is smaller than a threshold. In this framework, the authors consider several variants of SGD, including distributed variant and SVRG, for each of which the authors study the expected number of iterations for a prescribed accuracy under an assumption between the training and validation set.

While the use of a validation function is useful for early stopping, it introduces additional cost.

While bounds on the expected number of iterations are derived for several variants of SGD, it seems that most arguments are adapted from the existing analysis to take into account the validation function.

In Corollary 3.4 and Corollary 3.5, the bound is an increasing function of m. This suggests that the best choice would be m=1. However, in this case, one needs to calculate the gradient of the validation function at each iteration, which may wastes a lot of computation.

The authors consider constant step sizes. In practice, step sizes are often decreasing along the optimization. Can the analysis be extended to cover the case with decreasing step sizes?

In eq (30), there is a missing factor of 2.

There is a required $\epsilon>G62d_1(\mu_V,\mu_T)^2$ in the results. Therefore, to achieve a high accuracy we need $d_1(\mu_T,\mu_T)$ to be small. How many numbers of sample size to make $d_1(\mu_V,\mu_T)$ small? This has an influence on the computational cost.


----------------------
After rebuttal:

The authors do not respond. I would like to keep my original score.

**Experience Assessment:**

I have published one or two papers in this area.

**Review Assessment: Checking Correctness Of Derivations And Theory:**

I did not assess the derivations or theory.

**Review Assessment: Checking Correctness Of Experiments:**

I did not assess the experiments.

**Review Assessment: Thoroughness In Paper Reading:**

I read the paper at least twice and used my best judgement in assessing the paper.

---

### Official Review · AnonReviewer3 · 2019-10-28
**Official Blind Review #3**

**Rating:** 3

**Review:**

This paper proposes an optimization approach in which the optimizer computes the gradient on a given function yet uses another to decide a stopping time. Conceptually those functions are empirical errors on train and validation folds in the most common setting, although the authors seem to use other settings later in the paper to consider decentralized optimization schemes. The authors introduce a bound on the Wasserstein distance between the train and validation distributions in their analysis which plays a crucial role in their results. The authors use these results to motivate variants of existing optimization algorithms.

The paper is interesting but its message is a bit blurred to me. I had trouble pinpointing one main contribution, since the paper is split as theory (with some results) and a collection of slightly modified SGD type algorithms that are now impacted by this "gradient somewhere / monitor progress elsewhere". The theoretical results are worth reading and the idea appealing.

The paper also requires a *lot* of polishing. It has been sloppily written. For these reasons I am inclined to reject the paper and encourage the authors to improve their draft with a better formulation.


Minor comments:
- I have found the "main contributions" paragraph to be poorly phrased. Since the authors only monitor the validation loss and not the training loss, I do not think this falls into the "standard" definition of early stopping.
- please use citet and citep consistently.
- please add labels to figures and format them properly (e.g. SSGD on p.6)
- unsure about the format used to display f_T(x_t) in p.6
- Villani 2008 has over 900 pages. any page in particular?
- Assumption 2.3 requires significantly more work... All bounds scale as G^2 (e.g. eq.9, 10,11), therefore an idea of what G's impact on the analysis sounds crucial. In Example 2.4 the authors start working out an example, but wouldn't it be more interesting to carry that out completely, e.g. for the KL? What kind of bound would that result in?
- I find it disturbing that important comments on some of the crucial quantities (such as descent direction Eq.3) are left out of the algorithmic box... This defeats the purpose of having an algorithmic box.


**Experience Assessment:**

I have read many papers in this area.

**Review Assessment: Checking Correctness Of Derivations And Theory:**

I assessed the sensibility of the derivations and theory.

**Review Assessment: Checking Correctness Of Experiments:**

I assessed the sensibility of the experiments.

**Review Assessment: Thoroughness In Paper Reading:**

I read the paper at least twice and used my best judgement in assessing the paper.

---

### Decision · Program_Chairs · 2019-12-19

**Decision:**

Reject

**Comment:**

The authors made no response to reviewers. Based on current reviews, the paper is suggested a rejection as majority.